# Improving the efficiency of adaptive management methods in multiple fishways using environmental DNA

**Masahiko Nakai[1], Taku Masumoto[2]\*, Takashi Asaeda[3], Mizanur Rahman[3]**

1 Japan International Consultants for Transportation Co., Ltd, Tokyo, Japan, 2 Energy Planning Department, East Japan Railway Company, Tokyo, Japan, 3 Saitama University, Saitama, Japan

\* t-masumoto@jreast.co.jp

**Data Availability Statement:** All relevant data are within the manuscript.

**Funding:** The authors received no specific funding for this work.

## Abstract

Dams and weirs impede the continuity of rivers and transit of migratory fish. To overcome this obstacle, fishways are installed worldwide; however, management after installation is important. The Miyanaka Intake Dam has three fish ladders with different flow velocities and discharges and has been under adaptive management since 2012. Fish catch surveys, conducted as an adaptive management strategy, place a heavy burden on fish. Furthermore, a large number of investigators must be mobilized during the 30-day investigation period. Thus, a monitoring method using environmental DNA that exerts no burden on fish and requires only a few surveyors (to obtain water samples) and an in-house analyst was devised; however, its implementation in a fishway away from the point of analysis and with limited flow space and its effective water sampling frequency have not been reported. Therefore, in 2019, we started a trial aiming to evaluate the methods and application conditions of environmental DNA surveys for the continuous and long-term monitoring of various fish fauna upstream and downstream of the Miyanaka Intake Dam. To evaluate the fish fauna, the results of an environmental DNA survey (metabarcoding method) for 2019 to 2022 were compared to those of a catch survey in the fishway from 2012 to 2022. The results confirmed the use of environmental DNA surveys in evaluating the contribution of fishways to biodiversity under certain conditions and introduced a novel method for sample collection.

## Introduction

Dams and weirs disrupt river continuity and impede migratory fish movement [1,2]. The risks from habitat degradation due to dams are possibly much higher than those from climate change [3–10]. The focus of conservation has gradually shifted from only fish, considered as economic resources in Europe and the United States, to overall biodiversity, which includes a variety of fish [11]. Thus, the protection of biodiversity upstream and downstream of dams is critical, and such work is based on assessments of factors that include river gradients, tributary confluence, riverbank environments, and downstream conditions [12,13]. Structures that

**Competing interests:** The authors have declared that no competing interests exist.

cross rivers interrupt the continuity of habitats for organisms that must travel upstream and downstream. Although fishways have been installed worldwide to remove these obstacles, management following fishway installation is critical and should consider distribution patterns and extinction risks [14–16].

The monitoring required for conventional management is performed using net capture and expert analysis; however, these methods are costly and require considerable labor [17,18]. In areas where local survey data are lacking, complementary biological assessments may be conducted using data from species distribution models that are readily available elsewhere [19]. Segments originating from various types of vertebrates, such as fish and amphibians, sequenced via normal PCR amplification contain sufficient nucleotide differences to distinguish between animal species [20]. Surveys using environmental DNA present high cost effectiveness and biodiversity assessment standardization [21]; thus, they are considered suitable for understanding the diversity of fish and reduce the required time and expenses [22–30]. Moreover, because of species extinctions, ecosystems must be assessed before and after structure formation. Therefore, environmental DNA could represent a sensitive and powerful tool for monitoring such interruptions in river continuity [31–33].

In 2008, the first successful identification of organisms using environmental DNA occurred when the presence and density of the alien species *Rana catesbeiana* were inferred by extracting environmental DNA from pond water [34]. Subsequently, the development of metabarcoding technology, which detects multiple types of DNA simultaneously, progressed in Japan. Attempts were made to obtain a more complete picture of fish species by applying noninvasive methods to detect target alien species. In 2011, "all types of DNA from aquarium water" and "four types of DNA from wild water" were successfully detected using environmental DNA for the first time [35]. This method is more sensitive than capture surveys [36,37]; therefore, it is more effective when targeting rare organisms [38–42]. Environmental DNA has been successfully used in ocean studies and is becoming a standard method in this field to enhance the study of ecosystems. Acoustic sonar, which has an established track record in sea environments, can also be applied in rivers to detect deep rivers and salmon, which are strong swimmers [43]. However, because of the uncertainty of recognition by acoustic sonar, the environmental DNA technology approach, which has been validated in the ocean, may also become the standard for rivers [44–46]. Optimal filtration contributes to ensuring optimal environmental DNA to maximize the detection probability in rare species habitats [47]. Moreover, compared to aqueous environmental DNA, sedimentary environmental DNA useful for benthic inference may be confounded due to resuspension and transport [48].

Analyses using environmental DNA have been performed for surveys of a wide range of target areas, including remote regions, irrespective of the classification of organisms. In many inaccessible locations, this method requires multiple data sources to establish the potential presence or absence of species [49]. The potential of environmental DNA has been strengthened by its ability to reveal unrecorded biodiversity components and human influences [50]. This method is widely used for analyses of organisms in water and soil [51]. Discussions about advances in environmental DNA indicate that this method will have wide-ranging applications for information processed based on conventional morphological classifications [52]. Using a non-idealistic environmental DNA barcode data approach facilitates access to new taxa [53]. Metabarcoding for biodiversity assessment provides a rapid and inexpensive alternative that does not require taxonomic expertise; in addition, instead of routine identification, researchers can focus on important aspects, such as species habitats [54]. The development of metabarcoding alleviates some of the challenges in morphological identification of bioindicator taxa [55]. The remarkable advances and developments in environmental DNA sequencing have eliminated the need for DNA amplification, and collections of DNA with standardized barcodes

have created a comprehensive taxonomic reference [56]. Therefore, prior to its practical use, environmental DNA underwent a number of experimental verifications [57]. Even in the absence of a complete database, elusive patterns of biodiversity may be revealed in environmentally heterogeneous and biologically diverse regions using environmental DNA metabarcoding [58]. Seasonally different activity patterns in different species of the same genus affect the probability of environmental DNA detection. Natural history information can therefore guide monitoring plans [59].

Organisms that cannot be directly observed are also detected using this method because environmental DNA carries the molecular signatures of a species [29], which has been confirmed in several studies [60,61]. However, an understanding of the history of environmental DNA analysis by investigators and researchers is required [62]. Species can be clearly detected in areas where they are not actually present [63]. Controlling false positives remains a major challenge in environmental DNA analysis [64]. Misidentification hinders the reliability of DNA-based assessments of biodiversity [65]. A balanced discussion with consistent communication, controls, and limits of detection to clarify false positives is important for resolving misconceptions about false positives [66]. Analyzing more filters can reduce the risk of false negatives when individuals live in low densities [67]. When used appropriately, this method can be used to formulate plans for managing ecosystem conservation [68–70]. Moreover, this method allows for a better understanding of the relationship between changes due to natural or artificial factors and fish populations [71]; therefore, this approach is recommended [72].

The diversity of fish based on environmental DNA in a space that presents limited flow and occurs a certain distance from the point of analysis, such as in a fishway, is poorly understood because of the limited number of reported surveys [73]. Therefore, in this study, we focused on developing a continuous, long-term, and efficient method for understanding the diverse fish fauna within fishway groups and elucidating the optimal water sampling frequency for environmental DNA research that can corroborate the results of real fish capture research. The Miyanaka Intake Dam has three fishways (ice-harbor-, stair-, and rock-ramp-type), each of which maintains its own independent environment. The most significant feature of the Miyanaka Dam is the difference in its flow velocity settings [74]. Specifically, the flow velocity in the ice-harbor-type fishway is two-fold that of the rock-ramp-type fishway [75]. During the fishway improvement project, which targeted 16 types of fish species, the fish targeted for each type of fishway were determined. The ice-harbor-type fishway is intended for use by *Oncorhynchus keta* and *Oncorhynchus masou*, the stair-type fishway is intended for *Plecoglossus altivelis* and *Tribolodon hakonensis*, and the rock-ramp-type fishway is intended for bottom-dwelling type fish and small fish with poor swimming ability. Considering studies determining the overflow depth of the bulkhead suitable for the target fish, an upstream environment with different current speeds was created [76]. Subsequently, it was confirmed that the flow velocity within each fishway remained below the rush speed of the targeted fish species. Monitoring studies conducted since 2012 have shown that fish preferentially select certain fishways based on their biological characteristics [75]. Therefore, the fish fauna varies among fishways. We conducted fish catch surveys since 2012 by setting traps at the upstream end of the fishway and identified 37 species of fish. Monitoring surveys based on captures have confirmed that the improved fishway is fully functioning [76]. However, it became clear that catching surveys are a heavy burden on fish [77,78] and that the number of individuals caught in fishways fluctuates with various factors, such as river water temperature, turbidity, and flow conditions due to flooding. A total of 180 investigators surveyed the fishway for one month. Specifically, six investigators each day retrieved catch baskets from the traps in the fishway once every hour and checked the types of fish eight times a day. This analysis has been conducted since 2012, and it is performed in June, when *P. altivelis* and *T. hakonensis*, which attract particular

attention in this river and represent small fish, swim upstream. Surveys have focused not only on the number of species but also on the number of individuals. However, the populations of migratory fish such as *Plecoglossus altivelis* and *Oncorhynchus keta* fluctuate owing to not only downstream environmental conditions during their upstream migration but also autumn floods during the spawning season and changes in seawater temperature. In another survey on salmon conducted by the Ministry of Land, Infrastructure, Transport and Tourism in the same fishway, the impact of natural fluctuations, such as the number of salmon migrating to Niigata Prefecture, was evaluated as one of the factors behind the changes in salmon populations. Regarding the relationship between the number of individuals counted and environmental DNA, a study showed that the environmental DNA concentration was higher in ponds with a larger number of individuals estimated via existing methods [79]. A correlation was confirmed between individual density and biomass estimated by the capture survey in the upstream section from the water sampling point and environmental DNA concentration [80]. However, in other studies, environmental DNA concentrations were not correlated with biomass or population [81]. One of the reasons for this could be that all measured samples were below the quantification limit for the target DNA sequence, such as in the case of American crayfish [82]. In addition, in an experiment in which the dilution of environmental DNA released in a river was measured as it flowed downstream, the concentration of DNA decreased in the downstream direction when the flow velocity was low. However, when the flow rate was high, the same DNA concentration was observed both downstream and upstream [83]. In addition, the following points were evaluated in order to develop a new research method: 1) effects when the number of migratory individuals increased or decreased significantly, 2) types of fish that migrated upstream in different fishways with different current velocities and discharges, and 3) presence or absence of changes in the fish fauna over a long period of time. Based on our findings, a detailed analysis of the population numbers of *P. altivelis* and *T. hakonensis* was performed in another study that drew from these survey results. Furthermore, a modification in a survey policy that focuses on changes in fish fauna was considered. We hypothesized that surveys using environmental DNA would not only reduce the burden on fish and surveyors but also contribute to the efficient and economical realization of medium- to long-term surveys by appropriately setting the number of places and times to obtain water in a day.

Therefore, in 2019, a study was conducted to identify fish fauna using environmental DNA analysis. We compared the results of the metabarcoding method with those of capture surveys for fish fauna around the Miyanaka Intake Dam. Masumoto et al. [76] highlighted the most distinctive feature of the Miyanaka Intake Dam: the fish fauna varies across its three fishways. This offers a fine-grained system to benchmark environmental DNA against capture surveys. It was crucial to clarify the fish species that did not overlap between the two surveys. Next, a study was conducted using environmental DNA analysis to understand the utilization of these fish passages by different fish species.

Environmental DNA analysis (metabarcoding method) was performed for each fishway, and the findings were compared with the results of the corresponding fish catch surveys [84–88]. The metabarcoding results for 2021 and 2022 from the three different types of fishways (ice-harbor-type, stair-type, and rock-ramp-type fishways) were compared to evaluate whether the status of fish can be understood based on fish passages through an environmental DNA survey. Specifically, the effectiveness of determining the fishway usage status using an environmental DNA survey was examined by focusing on reproducibility and comparing the characteristics of the fish fauna in terms of the amount of environmental DNA between fish passages. In addition, changes in the number of fish caught were compared with changes in environmental DNA to understand the use of fishways through environmental DNA surveys. The

amount of DNA corresponding to fish that showed migration status changes between 2021 and 2022 was evaluated under the assumption that if the amount of environmental DNA in a fishway reflects the use of the fishway by fish, the amount of DNA for species with changes in the number of catches from 2021 to 2022 will also show corresponding changes.

The effort required for water sampling was less than that for the fish-catching survey, and the cost of the environmental DNA survey was lower, even after accounting for the cost of reagents and analysis [89]. Here, we aimed to obtain results that could be substituted for continuous, long-term, and efficient evaluation of various fish fauna in the fishway without imposing a burden on the fish.

## Materials and methods

### Survey points, timing, and number of water samples

The target fishways are located on the right bank of the Miyanaka Intake Dam (37˚3'58.445"N, 138˚41'50.321"E) at 134 km from the mouth of the Shinano River [76]. There were four collection points for environmental DNA analysis in 2022 (Fig 1): st1 was located at the exit of the fishways (upstream) and st2–4 were located at the entrance (downstream) of each of the three fishways (ice-harbor-type, stair-type, and rock-ramp-type). st1 was set as the control point for assessing the environmental DNA flowing into the fishway, and st2–4 were used to assess the effect of fish existing in the fishway. To prevent differences in the environmental DNA flowing into each fishway, st1 was set at a point before the river was divided into the three fishways. st2–4 were set slightly upstream of the submerged area and established to evaluate fish

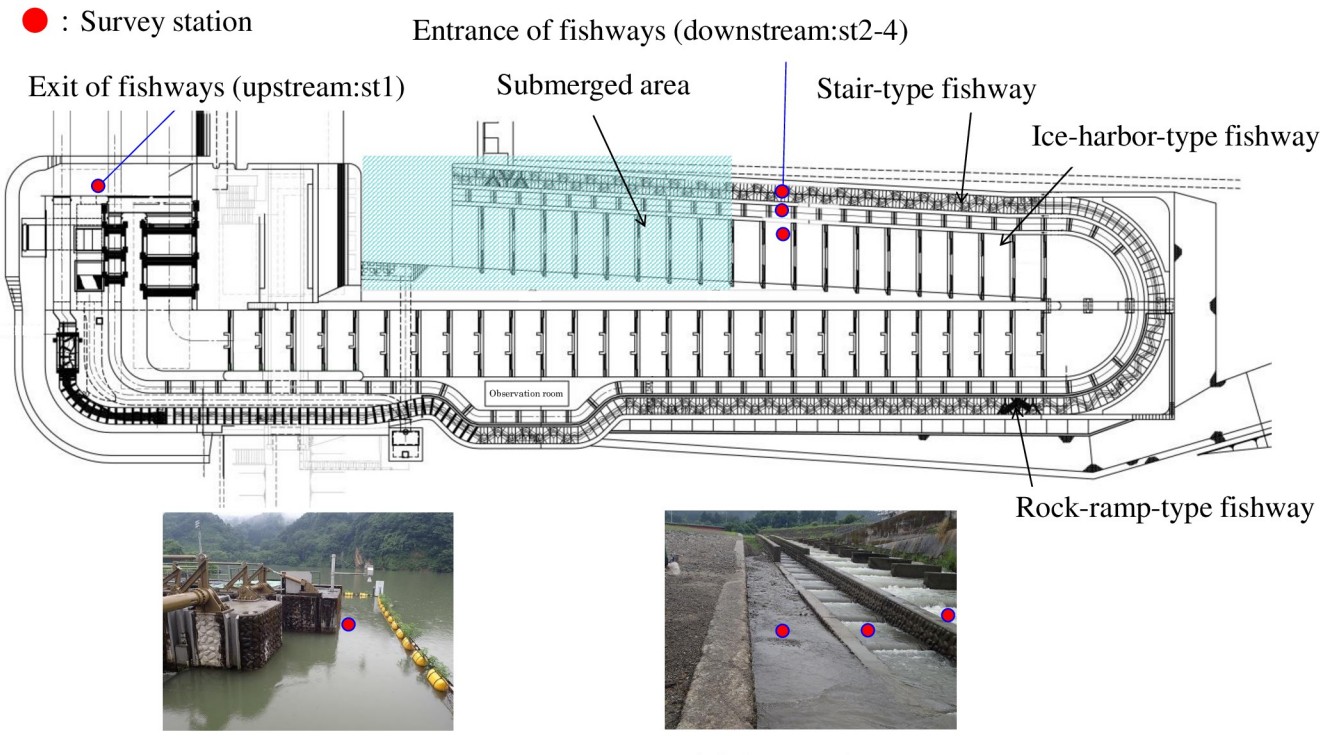

**Fig 1. Survey station in the fishways of Miyanaka Intake Dam.** Three types of fishways with different flow velocities and water depths are installed on the right bank side.

swimming up the fishway or staying in the fishway based on the difference in concentration of environmental DNA between st1 and st2–4. There were two water sampling points in 2019 and 2020 (st1 and st2) and four in 2021 and 2022 (st1 and st2–4).

Samples used for environmental DNA analysis were collected during the same period as that of the comparative capture survey (29-day period from June 6 to July 4). Water sampling was excluded on days when fish were not expected to migrate due to flooding. The capture survey was carried out every hour from 9:00 to 17:00 each day; water sampling was carried out at the same times.

As a result, 360 water samples for environmental DNA analysis were collected in 2019 (two sites × 9 times/day × 20 days), 408 were collected in 2020 (two sites × 9 times/day × 20 days + two sites × 2 times/day × 12 days), 972 were collected in 2021 (four sites × 9 times/day × 27 days), and 936 were collected in 2022 (four sites × 9 times/day × 26 days). To determine the frequency (time interval and date interval) sufficient to replace the capture survey with an environmental DNA survey, samples were selected for analysis based on the results of the capture survey.

## Survey methods using environmental DNA

**Collection and preparation of samples for environmental DNA.** Water sampling and analysis were carried out according to the Environmental DNA Sampling and Experiment Manual Version 2.1 (published on April 25, 2019, environmental DNA Methods Standardization Committee). A bucket with an attached string, water sampler with a handle, and measuring cup were used according to the sampling environment. These instruments were decontaminated with chlorine bleach before and after use to prevent false positives. New gloves were used when handling water sampling equipment and water sampling pads and performing filtration work. In st1 where the height difference of the water surface and water sampling point was approximately 2 m, a bucket with a string was used to sample surface water. In st2 and st3 where surface water was sampled across a rock-ramp-type fishway, a water sampler with a handle was used. Surface water was used as the sample because the bottom water and surface water in a 2 m deep pool mix as they flow down more than 200 m. In st4, where the environment was normal, a measuring cup was used to collect water samples near the center. One liter of water was collected for each sample (Fig 2). Environmental DNA research workers avoided contact with fish capture research workers who were working in the same fishway management yard to prevent false positives.

The collected samples were injected into a disposable water collection bag. To protect against DNA degradation, which can cause false negatives, 1 ml of benzalkonium chloride 10% w/v aqueous solution (commercial product chloride: OSVAN) was added for every 1 L of water. Following the temporary storage under cold conditions, the samples were filtered through glass fiber filters (mean pore size, 0.7 μm) on site within a few hours [90,91]. A filtration device developed for this study that is capable of filtering 10 samples at the same time was temporarily installed in a car parked at the management yard next to the fishway where the water was sampled. The filtration device used a 100 V power source and thus could be operated using car batteries. The time required for one filtration was approximately 15 to 30 min. Because DNA degrades over time, short-term filtration is desirable. However, increasing the pump capacity may cause the filter paper to tear. A time of 15–30 min is appropriate because the samples were collected once every hour and a pump type was used instead of free-fall. The filtered sample (filter paper) was folded in half, with the filter side toward the interior; it was shielded from light using aluminum foil, placed in a small plastic bag, and stored in a frozen

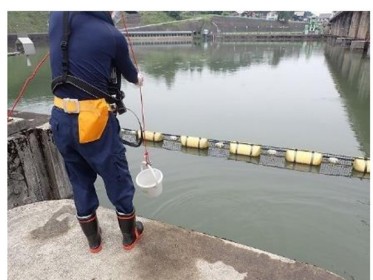

The bucket with a string in st1

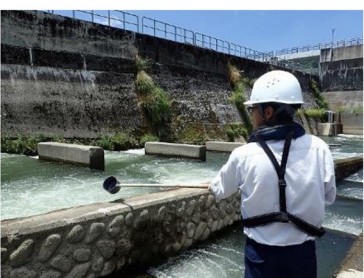

The water sampler with a handle in st2 and st3

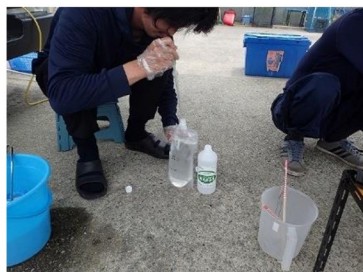

Addition of benzalkonium chloride

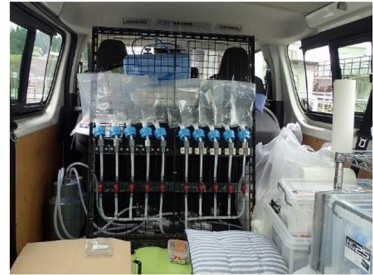

Filtration status by the filtration system built in the car

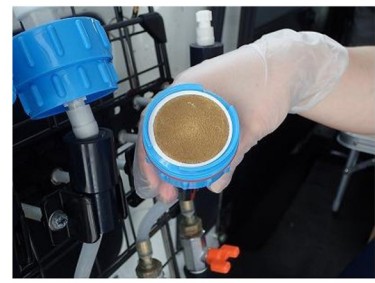

Condition of filter after filtering

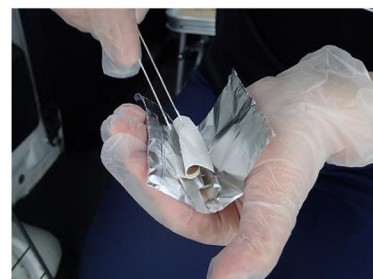

Light shielding with aluminum foil (bi-fold)

**Fig 2. Water sampling, developed in-vehicle unique filtration system, and storage of filter paper.**

state (below −20˚C) [92]. Specifically, the filter paper was stored in a portable freezer on site, and after one day of research was complete, it was stored in a stationary freezer indoors.

The survey required more than 900 water samples over 29 days for analysis based on the results of the capture survey. Filtration performed as soon as possible after sampling and cold storage contribute to the longer retention of analyzable concentrations of environmental DNA, as the concentration of environmental DNA in water samples declines rapidly over time [93,94]. Environmental DNA was successfully stored at room temperature (20˚C) for 2 weeks [95–98]. However, the Niigata Prefecture, where the water was sampled, and the Kyoto Prefecture, where the analysis was performed, are approximately 500 km apart. Completing the transport before the rapid DNA degradation in all water samples was difficult; therefore, a unique filtration system was built into the car taken to the water sampling point [99]. On-site filtration and cold storage (−20˚C) enabled efficient collection of a large number of samples.

**Selection of sample for analysis.** The water sampling frequency was adequately verified to ensure the development of environmental DNA surveys as an alternative to capture surveys. Catch surveys were conducted hourly from 9:00 to 17:00 during the survey period (June 6 to July 4) [76]. To improve the efficiency of the alternative survey, the sampling frequency (period and time) was determined based on the results of a trial conducted since 2019.

In this exhaustive analysis, samples for environmental DNA analysis were selected based on the results of fish catch surveys (the number of species and individuals confirmed at each time point), taking into consideration the migration status of *P. altivelis* and *T. hakonensis* representing the ice-harbor-type fishway, *Opsariichthys platypus* representing the stair-type fishway, and *Rhinogobius kurodai* representing the rock-ramp-type fishway. Six specimens (two survey points at st1 and st2) were analyzed in 2019, the first year of the survey, which covered 3 days.

Twelve specimens (similarly, two survey points) were analyzed in the fiscal year 2020, the second year, which covered 3 days (11:00 and 15:00, respectively). To verify whether it is possible to understand the difference in the migration situation among the three types of fishways (ice-harbor-type, stair-type, and rock-ramp-type), the number of specimens obtained in 2021 and 2022 was 16 (four survey points for st1 and st2–4). The number of species and individuals of fish caught tended to increase in the afternoon. Furthermore, there was no clear difference between the two fish fauna analysis results at 11:00 and 15:00. Therefore, from 2021, fish fauna analysis was performed using the results at 15:00. Water sampling was still done hourly in case the trend of variety and population of fish caught changed. Specimens for analysis were selected based on the capture results. Based on the results of the advanced capture survey, the number of fish species was not expected to change significantly over time. Therefore, the results obtained at 15:00 each day, when the number of the confirmed types was high, were used as the sample.

Samples were not collected immediately after a flood, when the water is often cloudy and DNA is washed out, because of the potential to induce false negatives. To further prevent false negatives, the water was not sampled during the spawning season of the dominant species *P. altivelis*, and the water sampling point was not a spawning site.

**Analysis of environmental DNA.** The analysis procedure was performed according to Environmental DNA Sampling and Experiment Manual Version 2.2 (published April 3, 2020, environmental DNA Methods Standardization Committee). An outline of each analysis method is presented below [100]. DNA metabarcoding is a next-generation sequencing-based technique for the analysis of environmental samples. It uses marker genes to characterize the species composition of whole communities [101,102].

During analysis to prevent false positives, the DNA extraction room was sufficiently spatially separated from PCR-related rooms in accordance with the manual. Two primers were designed to flank the gene region of interest. The MiFish primers [103] that can amplify the DNA of all Osteichthyes species and *P. altivelis*-specific and *Lethenteron* spp.-specific primers were mixed for the metabarcoding in fish fauna analysis. *P. altivelis* and *Lethenteron* spp. are difficult to detect using the MiFish method; therefore, specific primers were added to prevent false negatives. MiFish primers are a set of primers developed for fish fauna analysis; they amplify the DNA of all ichthyes. However, amplifying DNA from some species is difficult. Therefore, the basic set U, *Pseudoblennius percoides* Günther-specific U2, and Chondrichthyes-specific E-v2 were mixed and used. DNA was eluted (56°C, 30 min) from selected samples (filter paper with environmental DNA attached) using a buffer containing proteinase K. DNA was extracted and purified from the filter using the DNeasy Blood and Tissue Kit (Qiagen, Hilden, Germany). The extracted DNA was stored frozen at −30°C until analysis. DNA was purified using a filter with selective affinity to DNA.

*Comprehensive DNA analysis.* Metabarcoding (simultaneous detection of multiple species) to understand the fish fauna was carried out in three steps: 1) library creation, 2) next-generation sequencing (NGS), and 3) data analysis (matching with database).

1) Library creation

The library was constructed using a two-step PCR: (1) 1st PCR to amplify the target DNA sequence, which was repeated eight times according to the Environmental DNA Sampling and Experiment Manual Version 2.2, and (2) 2nd PCR to add a tag (marker) that allows the target sequence to be identified during the sequencing. The reaction mix consisted of KAPA HiFi HS ready mix (6.0 μL), Primer Mix (MiFish U, U2, E v2)–F/R (2.0 μL), sample DNA (2.0 μL), and water (2.0 μL). The cycling conditions for the 1st PCR are as follows: one cycle at 95°C for 180 s, 35 cycles at 98°C for 20 s, 65°C for 15 s, 72°C for 15 s, and one cycle at 72°C for 300 s. The

**Table 1. Sequences of primers used for PCR.**

| Primer name | Sequence information |
|---|---|
| MiFish U-F | TCGTCGGCAGCGTCAGATGTGTATAAGAGACAGNNNNNNGTCGGTAAAACTCGTGCCAGC |
| MiFish E-v2-F | TCGTCGGCAGCGTCAGATGTGTATAAGAGACAGNNNNNNRGTTGGTAAATCTCGTGCCAGC |
| MiFish U2-F | TCGTCGGCAGCGTCAGATGTGTATAAGAGACAGNNNNNNGCCGGTAAAACTCGTGCCAGC |
| Ayu-F | TCGTCGGCAGCGTCAGATGTGTATAAGAGACAGNNNNNNGTCGGTTAATCTCGTGCCAGC |
| Yatsume-F | TCGTCGGCAGCGTCAGATGTGTATAAGAGACAGNNNNNNGTCGGTTAATCTCGTGCCAGC |
| MiFish U-R | GTCTCGTGGGCTCGGAGATGTGTATAAGAGACAGNNNNNNCATAGTGGGGTATCTAATCCCAGTTTG |
| MiFish E-v2-R | GTCTCGTGGGCTCGGAGATGTGTATAAGAGACAGNNNNNNGCATAGTGGGGTATCTAATCCTAG |
| MiFish U2-R | GTCTCGTGGGCTCGGAGATGTGTATAAGAGACAGNNNNNNCATAGGAGGGTGTCTAATCCCCGT |

U is a primer suitable for fish in general, U2 is applicable to *Pseudoblennius percoides* Günther, and E-v2 is specific to cartilaginous fish. These primer sequences all refer to the Environmental DNA Sampling and Experiment Manual Version 2.2. Ayu is the primer specific for *P. altivelis*, and Yatsume is the primer specific to *Lethenteron spp*. Each primer was designed with reference to the sequence stored on the NCBI public DNA database (https://www.ncbi.nlm.nih.gov/genbank/). The forward primer (F) was designed to anneal to the 3' end of the region to be amplified, while the reverse primer (R) was designed to anneal to the 5' end of the region to be amplified. Note that the primer sequence of RNA synthesis proceeds in the 5'→3' direction.

conditions for the 2nd PCR are as follows: one cycle at 95˚C for 180 s, 10–12 cycles at 98˚C for 20 s, 72˚C for 15 s, and one cycle at 72˚C for 300 s. MiFish metabarcoding uses 2-step PCR, with 35 cycles as the standard for the first PCR (1st PCR). However, if the fish are less active, it should be increased to a maximum of 40 cycles to recover a sufficient amount of DNA according to the Environmental DNA Sampling and Experiment Manual Version 2.2. No more than 40 cycles should be run, as excessive cycles can cause false positives. PCR cycles were carefully set to avoid inappropriate amplification.

In this study, MiFish primers that specifically amplify fish DNA and a SimpliAmp thermal cycler (Thermo Fisher Scientific) were used. Table 1 lists the sequences of primers used for PCR.

2) NGS

NGS was performed for the 2nd PCR product (library). The prepared libraries were sequenced on the MiSeq platform using the MiSeq Reagent v2 Kit (Illumina, CA, USA) for 2 × 150 bp PE or 2 × 250 bp PE following the manufacturer's protocol. Sequence information and read counts for each operational taxonomic unit (OTU) were obtained by collating the DNA sequences into OTUs (sequence-based taxon units). An OTU was obtained by classifying the base sequences of multiple amplified DNA fragments using similarity as an index. The name of the genus was identified by matching the base sequence of each OTU with those in the database. Processing was performed by paired-end sequencing. The target had an average length of 170 bp. All data preprocessing and analyses of MiSeq raw reads were performed using PMiFish version 2.4.1 (MIT license Copyright 2020 rogotoh, https://github.com/rogotoh/PMiFish; Miya et al., 2015) and USEARCH v11 (Edgar 2010) based on the following steps. Both forward and reverse reads were merged using the fastq_mergepairs command. In this process, low-quality tail reads with a cutoff threshold set to a quality (Phred) score of 2, reads that were too short (<50 bp) after tail trimming, and paired reads with too many differences (>5 positions) in the alignment region were discarded. The fastx_truncate command was used to remove primer sequences from merged reads, and the fastq_filter command was used to quality-filter reads without primer sequences, remove low-quality reads with expected error rates >1%, and remove reads that were too short (<50 bp). The preprocessed reads were dereplicated using the fastx_uniques command, and all singletons, doubletons, and tripletons

were removed from the subsequent analysis following the recommendation of Edgar (2010). The dereplicated reads were denoised using the unoise3 command, and all putatively chimeric and erroneous sequences were removed from the subsequent amplicon sequence variant (ASV) assignment. Finally, the usearch_global command was used to align all processed reads with sequences in a database prepared from GenBank, MitoFish (http://mitofishaoriu-tokyoacjp/), and an in-house database and then to assign sequences with >80% identity to fish sequences (Miya et al., 2015).

3) Data analysis (matching with database)

Candidate species names for each OTU were obtained by comparing the sequence of each OTU obtained from the NGS analysis with sequences in the MitoFish database (MitoFish http://mitofish.aori.u-tokyo.ac.jp/). The species names were determined according to the guidance for freshwater fish survey methods using environmental DNA analysis technology [104]. Specifically, it was determined based on the candidate species names of each OTU and the existing fish fauna around the Miyanaka Intake Dam. When multiple candidate species existed and could not be distinguished from the existing fish fauna, the possible species were listed together (e.g., *Misgurnus anguillicaudatus* and *Paramisgurnus dabryanus*) and treated as one species for convenience.

**Conditions for analysis.** A quantitative comparison of the number of reads obtained using metabarcoding within the same sample for each fish was possible. However, a comparison between samples was not possible because 1) the amount of PCR inhibitors varied between samples; 2) the dilution ratios at the time of analysis varied between samples; and 3) the number of amplifications varied between the samples because the threshold would be reached for samples with high DNA concentrations before the predetermined number of amplifications. However, the water flowing into the three types of fishways in the Miyanaka Intake Dam is the same water from the dam reservoir upstream of the fishway exit (st1); therefore, the water quality was assumed to be the same. These problems were interpreted as follows in this study: 1) the concentrations of PCR inhibitors could be considered approximately equal; 2) the metabarcoding results could be converted to the number of reads per unit volume based on the dilution ratio of each sample; and 3) differences in the number of amplifications were difficult to correct for. However, the differences were not large because the DNA concentrations in the samples were adjusted to a certain extent by the dilution during the analysis. Considering the conditions of the analysis in this light, the number of reads for each fish species obtained through metabarcoding was assumed to reflect the differences between the samples to some extent. Therefore, the metabarcoding results were converted to per unit volumes and used for comparison.

**Handling of data in comparison studies.** The environmental DNA obtained at the st1 water sampling points was influenced by both the environmental DNA originating from the fish in the fishway and the environmental DNA flowing from upstream of the fishway. A combination of cell sedimentation, turbulence, and dilution effects showed that detectable levels of environmental DNA extended by a distance of 239.5 m regardless of the current [83]. The length of the ice-harbor-type fishway and stair-type fishway of the Miyanaka Intake Dam is approximately 210 m, the length of the stair-type fishway is approximately 240 m, and the length of the rock-ramp-type fishway is 260 m. Therefore, the environmental DNA at St2–4 in the large and small fishways was not assumed to reflect the presence of fish in the fishway. To understand the contribution of environmental DNA derived from fish that migrate up the fishway in this study, the amount of environmental DNA at the upstream point (st1) was subtracted from the amount of environmental DNA (number of reads) at each point (st2–4), and

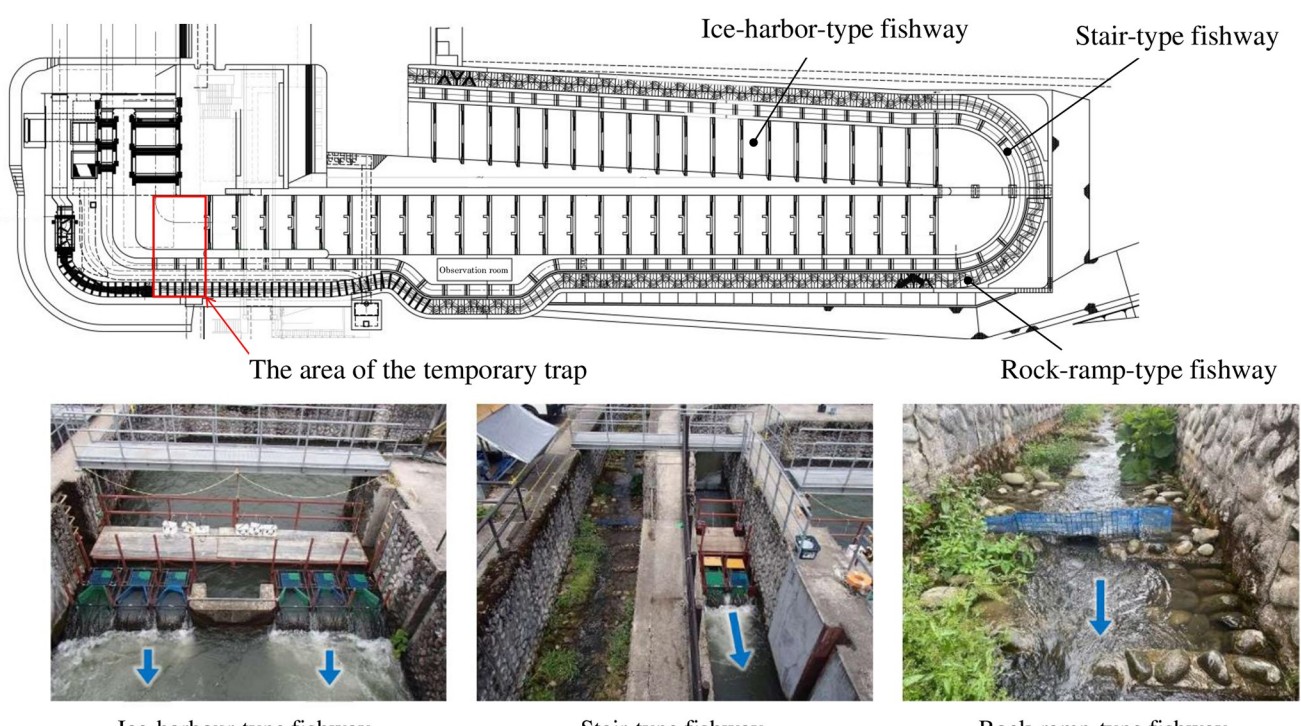

Fig 3. Position and state of the temporary trap for catching fish. Three traps were temporarily installed at the upstream ends of all fishways for one month each. Fish were caught in underwater net cages. The blue arrow indicates the flow direction.

the final value represented the environmental DNA derived from fish in the fishway. When the difference was negative, it was treated as 0.

## Method used for the fish catch survey

Comparative fish catch surveys have been conducted annually since 2012, and the rock-ramp-type fishway was constructed in 2012. A capture survey was conducted for 29 days from June 6 to July 4 using the method described by Masumoto et al. [76], with a temporarily installed trap as shown in Fig 3 in the fishways. The Governor of Niigata Prefecture granted a "Special Capturing Permit" (permit number: special No. 31) necessary for conducting the survey. The permit consisted of taking all animals for a specified period of time and reporting the results. A temporary net cage caught all the fish at the upstream end of each fishway. Catch baskets set at 9:00 caught fish every hour from 10:00 to 17:00. A dam upstream of the fishway received all fish after their length was recorded and photographed. Fig 4 presents the results of the capture surveys. To continuously confirm that *P. altivelis*, which represents small fish with a relatively weak swimming ability, reached upstream via the fishway, surveys continued to be carried out in June, during their migration season. Moreover, in order to verify the appropriate operation of the dam and fishway, their upstream status must be confirmed in the future, mainly in June.

## Narrowing down of samples for analysis

In this study, we considered reducing the frequency of water sampling to ensure continuous and efficient long-term monitoring of the fish fauna. For this reason, water was sampled at the same frequency as in the capture survey, although the samples to be analyzed were narrowed

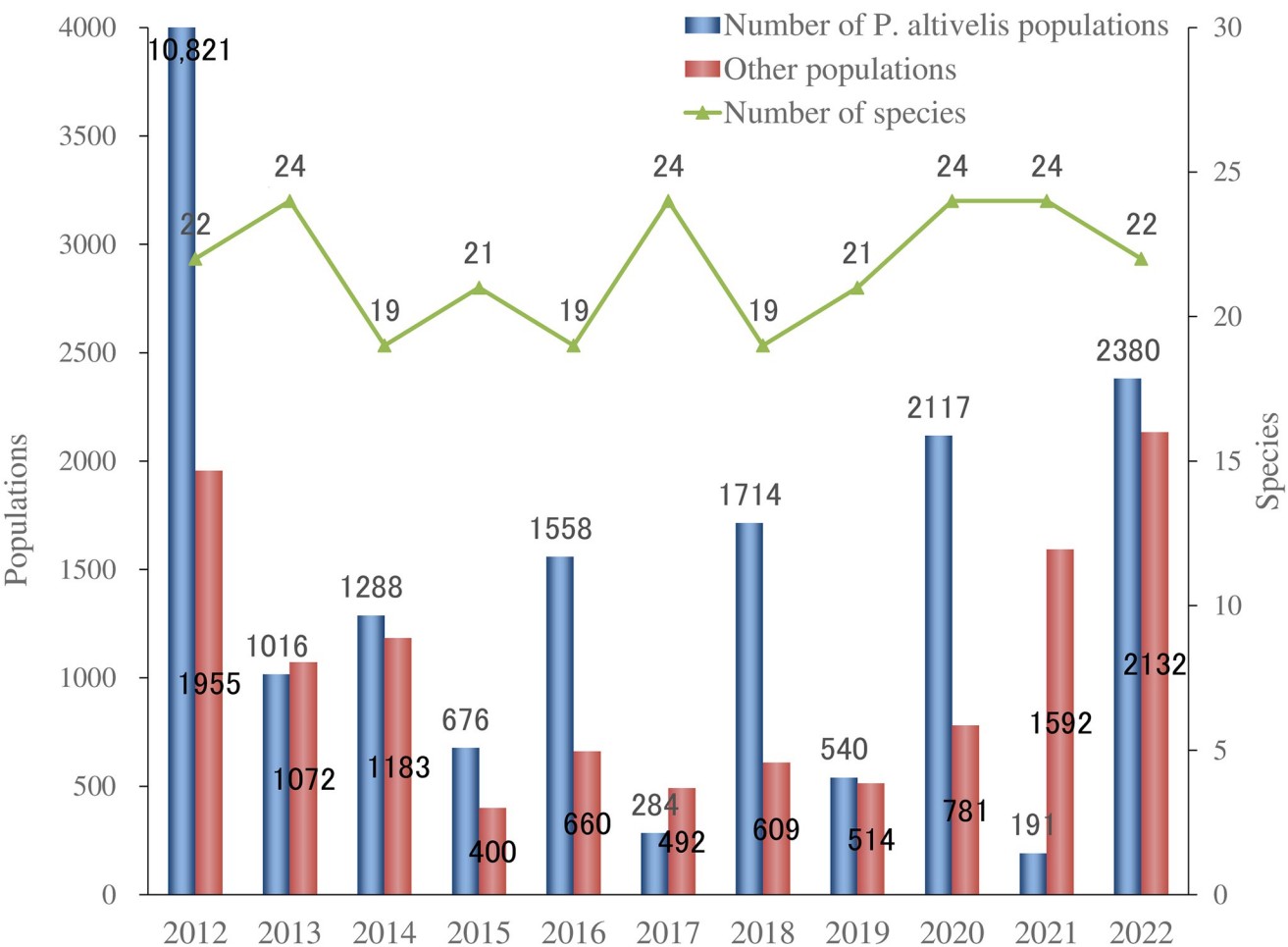

**Fig 4. Results obtained from catch surveys in fish passages (number of species, number of *P. altivelis* populations, and other populations).**

down based on the following considerations. The estimated results were evaluated by analyzing the samples at 15:00 on the target day (one sample per day), which was either a day when the number and species of fish migrating upstream on ice-harbor-type, stair-type, and rock-ramp-type fishways were high; a day with a large population of fish representative of each fishway (ice-harbor-type fishway: *P. altivelis*, *T. hakonensis*, stair-type fishway: *O. platypus*, and rock-ramp-type fishway: *R. kurodai*); or a day during the time period when the number of species and individuals was large on the days selected above. Rarely identified fish that did not affect the results of environmental DNA fish fauna analysis were not considered in the selection of days for analysis.

Environmental DNA surveys record fish species that inhabit the area because environmental DNA is measured in a range that includes the upstream side of the survey point for a short time. The DNA detection rate was negatively correlated with time, and the probability of undetectable DNA after 17 days was greater than 95% [105]. Studies have indicated that the effectiveness of biomonitoring using environmental DNA analysis requires preliminary verification of the DNA detection rate in an environment close to nature before conducting the analysis [106]. Therefore, we verified the period of time in which the number of confirmations by environmental DNA at a given time reflected the situation.

In the fishway fish catch survey, the species and number of fish caught in temporary traps at the upstream end of the fishway were investigated on an hourly basis. Therefore, the number of fish species recorded using environmental DNA may reflect the fish species upstream of the dam at a certain time. However, the number of fish species obtained from catch surveys reflects the situation in the fishway over a short period.

## Results

### Comparison of the metabarcoding and capture survey results

Table 2 shows the results of the metabarcoding from 2019 to 2022. The number of days of capture surveys and the number of specimens analyzed differed with the year of the survey; however, the number of species recorded in 2022 was the largest. The species observed in all samples also remained the same, and *Squalidus biwae* was newly recorded in all samples in 2022. *S. biwae* was also frequently caught in the fishway catch survey carried out in 2022. Therefore, this result was considered to reflect an increase in *S. biwae* population.

Fig 5 shows the results for two common locations (st1 and st2) from 2019 to 2022. The number of species confirmed by both the environmental DNA survey and the capture survey was 16–24. When including the number of species identified only by environmental DNA survey, the number of species was 19–35. Thus, the proportion of species confirmed by both environmental DNA survey and capture survey was 72.4% (65.7–84.2%) of the number of species confirmed using the environmental DNA survey (df = 6, t = 2.447, t<0.05). Species such as *Anguillidae* (one individual), *Cyprinus carpio* (one individual), *Carassius cuvieri* (zero individuals), *Lefua echigonia* (zero individuals), and *Gymnogobius urotaenia* (one individual) were identified only in the environmental DNA analysis; they were rare in the 4-year catch survey. However, species such as *Nipponocypris temminckii* (three individuals) and *Sarcocheilichthys variegatus microoculus* (two individuals) were confirmed only in the capture survey. A factor contributing to this false negative value was the low number of these fish that were caught.

A total of 37 species were identified in the fish catch surveys from 2012 to 2022, which was approximately the same as that recorded using environmental DNA surveys (Fig 6). Among the 37 species of fish identified using environmental DNA, *C. cuvieri* was the only fish species that had never been caught.

### Survey results for each fishway

**Effectiveness of evaluating fishway usage status through environmental DNA survey.** The composition of fish in each fishway, as indicated by the number of environmental DNA reads, differed with the survey year and date. However, two species, *M. anguillicaudatus* and *R. kurodai*, dominated the rock-ramp-type fishway; three species, *O. platypus*, *T. hakonensis*, and *P. altivelis*, dominated the ice-harbor-type fishway; and two species, *O. platypus* and *P. altivelis*, dominated the stair-type fishway.

Table 2. **Number of species detected by metabarcoding and captured by survey from 2019 to 2022.**

| Year | Specimens | Number of species detected by eDNA | Captured species | Fish confirmed in all samples |
|------|-----------|-----------------------------------|------------------|-------------------------------|
| 2019 | 6 | 20 | 21 | 4 |
| 2020 | 12 | 35 | 21 | 18 |
| 2021 | 16 | 31 (31) | 24 | 14 |
| 2022 | 16 | 36 (32) | 22 | 15 |

The numbers shown in parentheses (2021: 31 species, 2022: 32 species) are the number of species limited to st1 and st2 to match the conditions for 2019 and 2020.

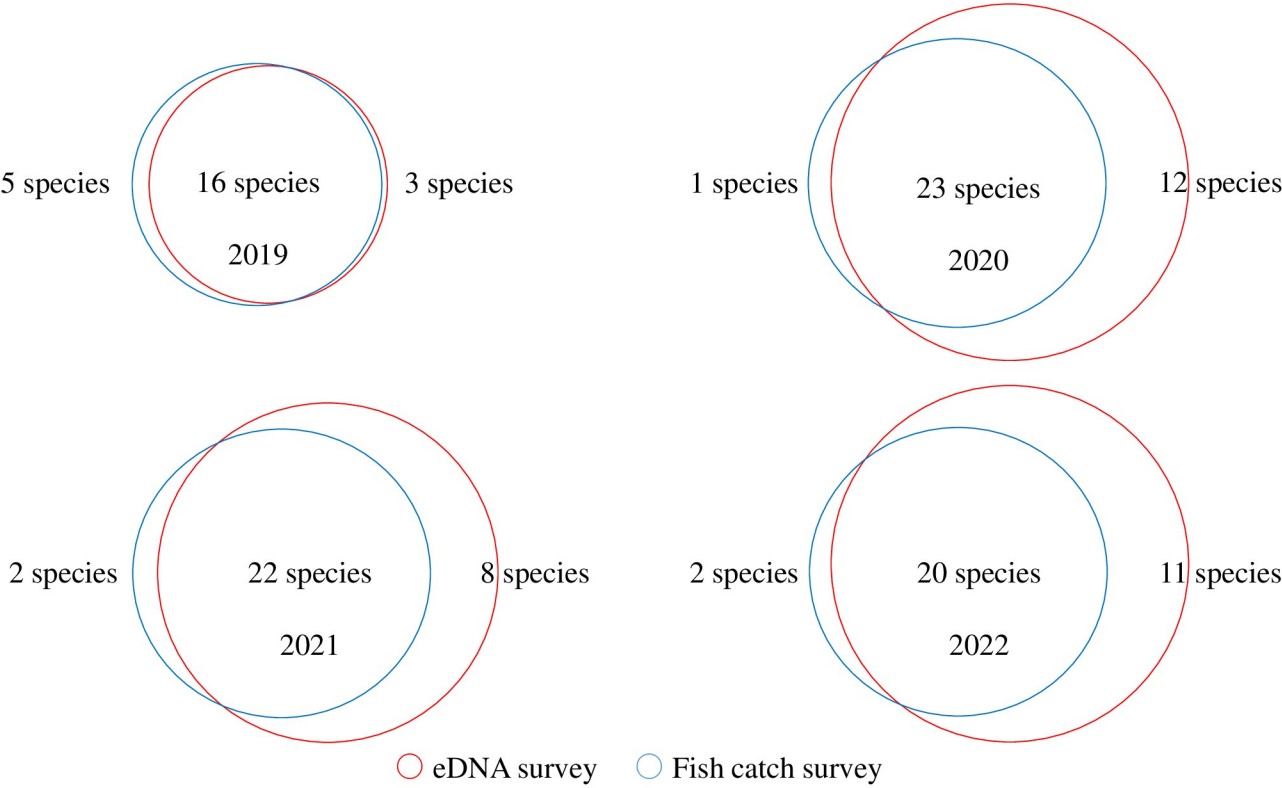

**Fig 5. Comparison of fish catch survey results and environmental DNA analysis results (2019–2022).**

In addition, the numbers of *Pelteobagrus nudiceps* and *S. biwae*, which ranked high in the stair-type fishway in 2022, increased significantly since 2021 (*P. nudiceps* increased from 9 individuals in 2021 to 125 individuals in 2022; *S. biwae* increased from 1 individual in 2021 to 122 individuals in 2022). The results of these environmental DNA surveys revealed a significant change in the number of individuals that migrated upstream.

**Effectiveness of understanding fishway usage status through environmental DNA survey.** Based on results of the catch surveys conducted in 2021 and 2022, the fish identified in the environmental DNA survey were grouped according to the fish representative of each fishway and their utilization. In Case 1, the percentage of read numbers for each fish species was examined. Fish that were predominantly caught in each fish passage, such as *O. platypus*, *T. hakonensis*, *P. altivelis*, *Cobitis biwae*, and *R. kurodai*, were distinguished as individual indicators. Fish other than these five species were prevalent in the ice-harbor-type, stair-type, and rock-ramp-type fishways. They were classified into groups 1 to 4 (Group 1: Fish species in which many individuals mainly use the ice-harbor-type fishway; Group 2: Fish species that mainly used the ice-harbor-type and stair-type fishways; Group 3: Fish species in which many individuals mainly used the rock-ramp-type fishway; Group 4: Other fish species with no clear difference in usage by fishway type). The ratio of DNA amount represented the average value for 4 days obtained through metabarcoding, and the ratio of capture survey results was calculated from the total number of captures during the survey period (29 days).

In Case 2, the fish of Group 2 and Group 3 were divided by lifestyle type (middle-level swimming type and bottom-dwelling type). In Group 2, *Squalidus chankaensis biwae*, *Sarcocheilichthys variegatus microoculus*, and *O. masou* were classified as middle-level swimming

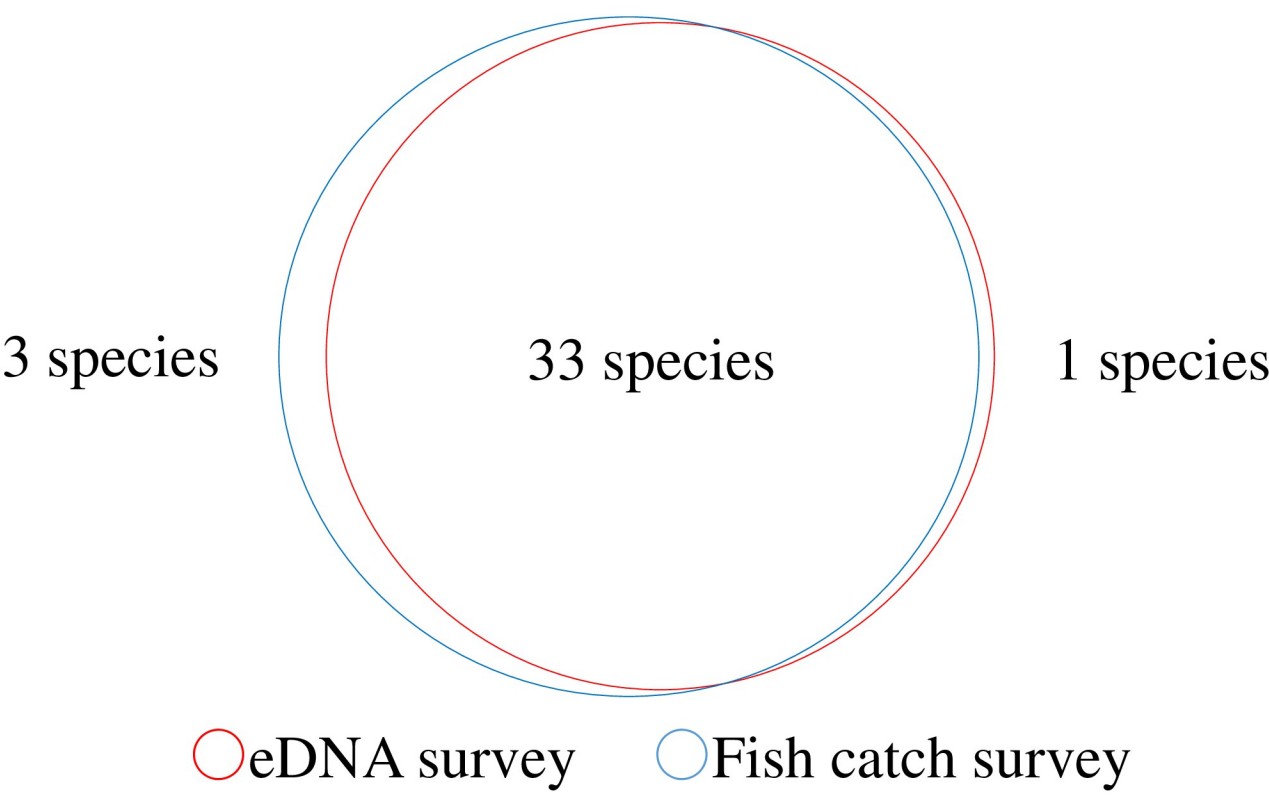

**Fig 6. Comparison of the number of fish species confirmed using environmental DNA surveys and capture surveys.** Environmental DNA survey results are the total for 2021 and 2022, and capture survey results are the total for 2012–2022. One species that was confirmed only through the environmental DNA survey is *C. cuvieri*, and the three species confirmed only through the capture survey are *Tanakia lanceolata* (past one individual), *Rhodeus ocellatus* (past five individuals), and *Tribolodon nakamurai* (past two individuals). The numbers of all of these fish were too small for confirmation.

types and *Cottus pollux*, *Pseudogobio esocinus esocinus*, and *Pseudobagrus nudiceps* were classified as bottom-dwelling types. In Group 3, *Rhynchocypris lagowskii steindachneri*, *Rhodeus ocellatus ocellatus*, *Pseudorasbora parva*, and *Gnathopogon elongatus elongatus* were classified as middle-level swimming types and *Liobagrus reini*, *Lethenteron reissneri*, *Misgurnus anguillicaudatus*, *Lefua echigonia*, and *Gymnogobius urotaenia* were classified as bottom-dwelling types. Furthermore, there was no bottom-dwelling type in Group 1, where many individuals mainly used the large fishway.

Fig 7(a) shows a comparison of the results of the environmental DNA survey and capture survey for each fishway. The fish fauna determined from the percentage of environmental DNA reads for each fishway was similar to the results of the capture survey. *T. hakonensis*, *P. altivelis*, and Group 1 fish constituted the main species in the ice-harbor-type fishway. *O. platypus*, *P. altivelis*, and Group 1 fish were the main species recorded in the stair-type fishways. In contrast, the ratios of *R. kurodai*, *C. biwae*, and Group 3 fish were high in the 2021 rock-ramp-type fishway. In addition, the number of swimming fish, such as *O. platypus* and Group 2 fish, increased in 2022. The discharge from the rock-ramp-type fishway during the 2022 catch survey was higher than that in 2021 owing to the operation of the fishway; many swimming fish were confirmed owing to this. Among the fish that migrated upstream through the rock-ramp-type fishway, bottom-dwelling type fish take time to reach the upstream end of the fishway. Therefore, their behavior was different from that of fish that migrated upstream through

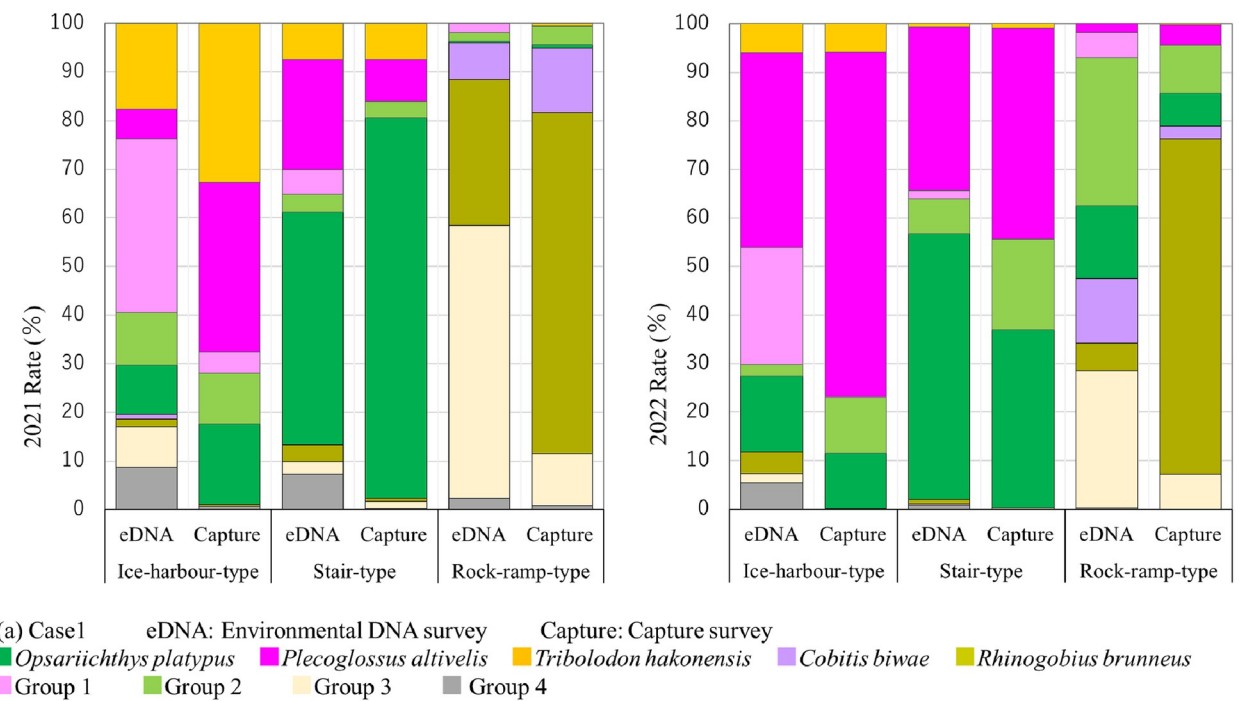

(a) Case1    eDNA: Environmental DNA survey    Capture: Capture survey
■ *Opsariichthys platypus*    ■ *Plecoglossus altivelis*    ■ *Tribolodon hakonensis*    ■ *Cobitis biwae*    ■ *Rhinogobius brunneus*
■ Group 1    ■ Group 2    ■ Group 3    ■ Group 4

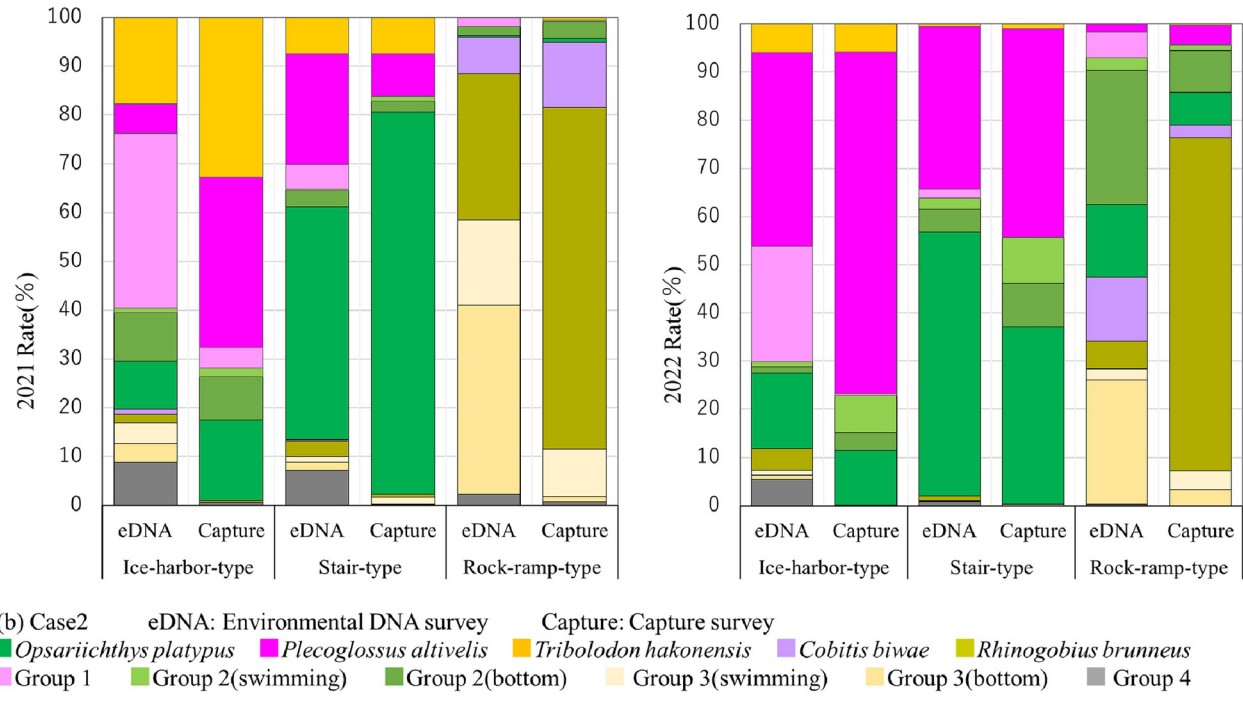

(b) Case2    eDNA: Environmental DNA survey    Capture: Capture survey
■ *Opsariichthys platypus*    ■ *Plecoglossus altivelis*    ■ *Tribolodon hakonensis*    ■ *Cobitis biwae*    ■ *Rhinogobius brunneus*
■ Group 1    ■ Group 2(swimming)    ■ Group 2(bottom)    ■ Group 3(swimming)    ■ Group 3(bottom)    ■ Group 4

**Fig 7. Comparison of environmental DNA survey results and capture survey results for each fishway.** (a) Case 1 and (b) Case 2.

the ice-harbor-type fishway and stair-type fishway. Thus, the differences between environmental DNA surveys and capture surveys may have reflected the behavior of these fish.

Fig 7(b) shows the results for Case 2. A clear difference was not observed in the proportion of environmental DNA reads (metabarcoding method), even when separated into middle-level swimming type and bottom-dwelling type, because the composition ratio of Group 2 and Group 3 was originally small in the ice-harbor-type fishway and stair-type fishway. In Group 3, the bottom-dwelling type was approximately twice as common as the middle-level swimming type in the rock-ramp-type fishway in 2021. In addition, the bottom-dwelling type accounted for the majority in both Group 2 and Group 3 in 2022.

The results of environmental DNA surveys captured such changes in the flow regime of the fishway. Therefore, identification of fish species in each fishway was possible by evaluating the read numbers of the major fish and the other groups of fish in the environmental DNA survey in each fishway.

## Division of analysis period to reflect capture results

The period during which the capture survey results were reflected in the environmental DNA results was identified because water sampling for environmental DNA and trap surveys was conducted hourly. To efficiently monitor the fish fauna over time and facilitate comparison with previous surveys, improvements were sought to enhance long-term survey efficiency. Evaluating the possibility of transitioning from a schedule of 30 days × 8 times/day to several days × 1 time/day, it was decided to solely use the samples collected at 15:00 as the environmental DNA sample. First, the results of the capture survey were divided into four survey periods: (1) that time, (2) that day, (3) 1 week before and after, and (4) 15 days before and after (targeting eight survey time slots: June 17, 21, 28, and 30 in 2021 and June 20, 22, 30, and July 2 in 2022, at 15:00 on every day). The relationship between the number of species confirmed by environmental DNA at 15:00 and the number of species captured at the four times shown above was arranged for each fishway. Fig 8 shows the results of verifying the natural state indicated by the environmental DNA at a specific time. As a result of verifying four periods (that time, that day, 1 week before and after (±3 days), and 15 days before and after (±7 days)) and the number of fish species caught in three fishways, the correlation coefficient (r) and decision coefficient ($R^2$) were large on that day (r = -0.587, $R^2$ = 0.345, p <0.05) and 1 week before and after (±3 days) (r = -0.442, $R^2$ = 0.196, p<0.05). Furthermore, when the sampling window was increased to 15 days before and after (±7 days), the r and $R^2$ values became smaller (r = -0.277, $R^2$ = 0.077, and p = 0.11). To investigate this factor, the correlation coefficient was verified in each fishway. The r values between the ice-harbor-type fishway and the stair-type fishway showed the same trend as those for the whole fishway (r>0.74, $R^2$>0.54, p<0.05), while the r values of the rock-ramp-type fishway showed a tendency different from those of the whole fishway (r<0.5, $R^2$<0.25, p<0.05). *P. altivelis*, *T. hakonensis*, and *O. platypus* were mainly caught owing to the fast current and deep water of the ice-harbor-type fishway and stair-type fishway. However, *C. biwae* and *R. kurodai* were mainly caught owing to the slow current and shallow depth of the rock-ramp-type fishway. Vegetation studies are being conducted on the rock-ramp-type fishway of the Miyanaka Intake Dam to enhance fish migration and habitat [107]. Therefore, *C. biwae* and *R. kurodai* migrate slowly and inhabit the rock-ramp-type fishway; accordingly, the results for the other two fishways are different [75]. As the number of species actually caught increased, many of them included fish with extremely low population numbers. Negative correlations occurred because of increased types not detected in environmental DNA (false negatives). Therefore, we focused on the five species of fish typically caught in the fishway of the Miyanaka Intake Dam and continued our verification.

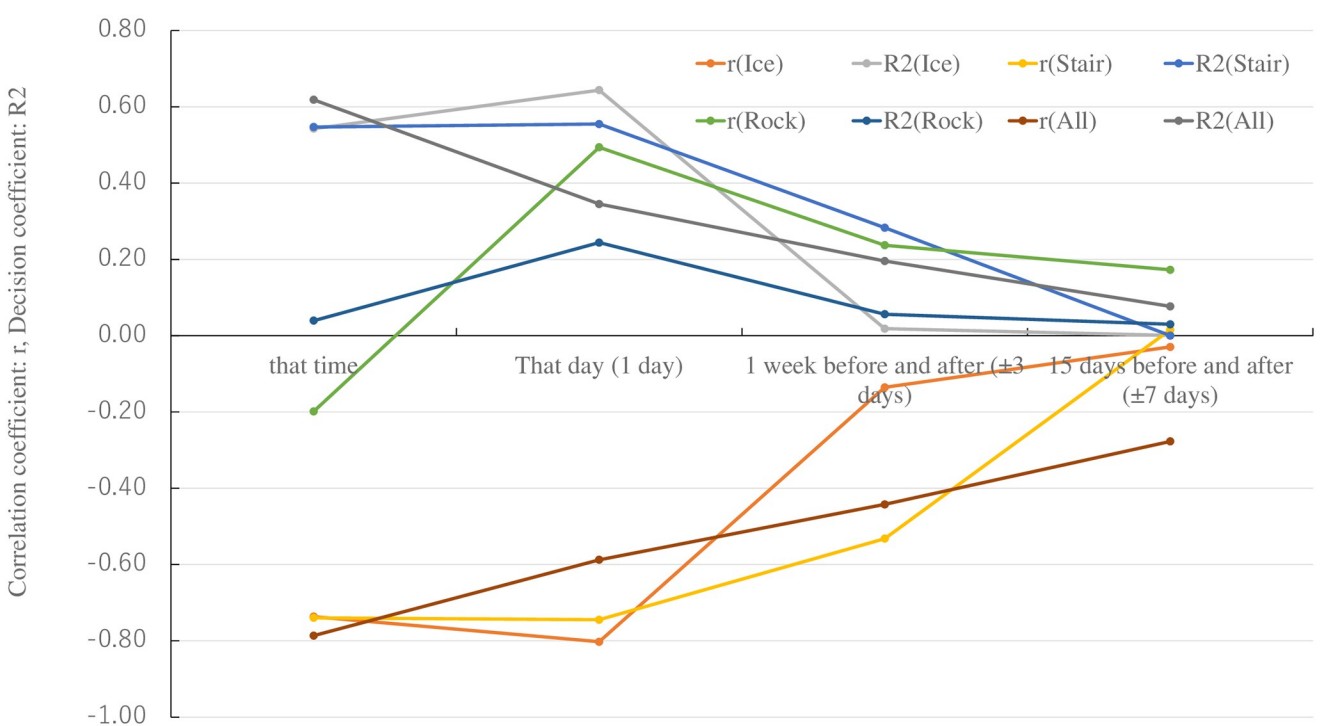

**Fig 8. Verification of natural conditions indicated by environmental DNA at a specific time targeting four periods and number of fish species caught in each fishway.** (Ice: ice-harbor-type fishway; Stair: stair-type fishway; and Rock: rock-ramp-type fishway).

Subsequently, a more detailed analysis was performed. The most frequently caught species, *O. platypus*, *T. hakonensis*, *P. altivelis*, *C. biwae*, and *R. kurodai*, were sorted. Fig 9 shows the correlation between the number of fish and environmental DNA reads for each survey period segment. Most *O. platypus* (approximately 82%) individuals used the low-flow stair-type fishways, and approximately 16% utilized the high-flow ice-harbor-type fishways. *O. platypus* has a short body length and low charging speed. Therefore, it is possible that a large amount of environmental DNA from *O. platypus* remained in the fishway because it required a long time to swim through the fishway. *O. platypus* was confirmed using environmental DNA in all survey period divisions; it was used as an indicator. Therefore, there was a significant correlation (r>0.4, especially a strong correlation for 15 days before and after; r>0.7, p<0.05, for 1 week before and after) in all survey periods. Most (approximately 70%) of the *T. hakonensis* individuals utilized ice-harbor-type fishways with high flow, and approximately 29% utilized stair-type fishways with low flow. Many *T. hakonensis* individuals migrated in groups, and large *T. hakonensis* individuals were strongly motivated to migrate. These factors contribute to short-term mass migration; this tendency reduces the retention of *T. hakonensis* environmental DNA in the fishway during non-migratory periods. Therefore, *T. hakonensis*, used as an indicator, was not recorded in the environmental DNA survey during four of the eight survey periods. Therefore, there was a weak correlation (r<0.4, p>0.05) in all survey period divisions. Most of the *P. altivelis* (approximately 63%) individuals used the ice-harbor-type fishways with high flow, and approximately 37% utilized stair-type fishways with low flow. The body length of *P. altivelis* was approximately 0.1 m, and all those that migrated upstream were juveniles in search of a habitat to establish a territory. Therefore, the number of captured *P. altivelis* individuals was large, indicating that their motivation to move upstream was high. However, few

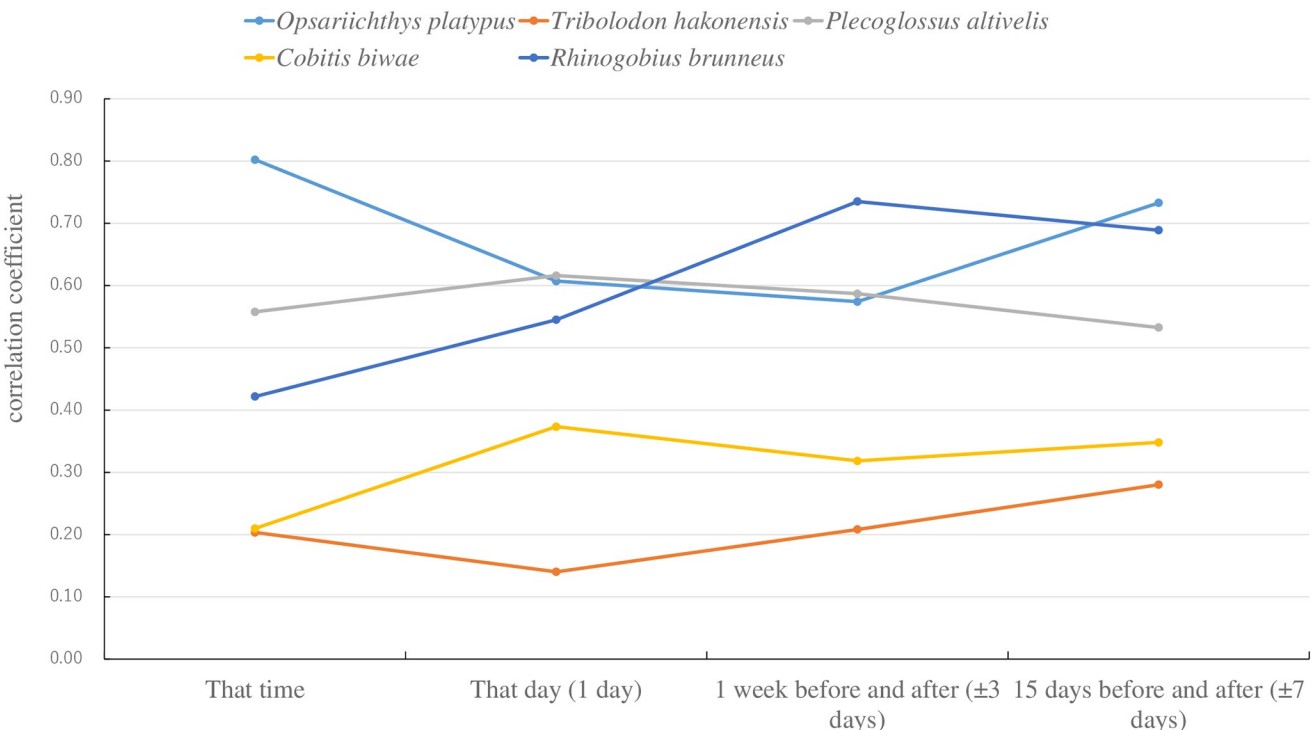

**Fig 9. Verification of natural conditions indicated by environmental DNA at a specific time targeting four periods and the five main types of fish caught.**

*P. altivelis* individuals (1%) were caught in the rock-ramp-type fishways with lower flows. *P. altivelis* was confirmed using environmental DNA analysis during all eight index sampling periods. Therefore, there was a significant correlation ($r>0.4$, $p>0.05$) in the survey period divisions in addition to that during the full survey period. *C. biwae*, a bottom-dwelling fish, was caught in rock-ramp-type fishways with low flow and still water. *C. biwae* was confirmed using environmental DNA analysis during all eight index survey periods. The total number of catches was as low as 76 individuals; thus, there was a weak correlation ($0.4>r>0.2$, $p<0.05$) in all survey period segments. *R. kurodai* not only migrated using the rock-ramp-type fishways but also lived in these fishways [72], and 637 individuals were captured. *R. kurodai* was confirmed using environmental DNA at all index times, indicating a significant correlation ($r>0.4$, especially $r>0.7$, for 1 week before and after, $p<0.05$) in all survey period segments.

The results of the capture surveys 1 week before and after and 15 days before and after were well reflected in the environmental DNA results. The ice-harbor-type fishway had a high discharge of approximately 1.637 m$^3$/s and a high velocity of 1.27–2.43 m/s. This suggests that the environmental DNA is probably diluted depending on the size of the fish species and speed of migration. The stair-type fishway had a flow rate of 0.133 m$^3$/s and a flow velocity of 0.87–1.05 m/s; therefore, fish with a strong swimming ability and bottom-dwelling fish do not use this fishway. The environmental DNA and capture survey results were relatively consistent because fish with a low swimming ability did not stay and took an appropriate amount of time to swim upstream. The rock-ramp-type fishway has a low flow rate (0.071 m$^3$/s) and low velocity (0.64 m/s). Small fish with a weak swimming ability took time to swim upstream. Many bottom-dwelling fish move slowly or live alone. The continuous release of environmental DNA from these fish suggests that the results of the environmental DNA and capture surveys did not

match depending on the type of fish. *C. biwae* was caught entirely from the rock-ramp-type fishways, and *R. kurodai* was partially caught from the ice-harbor- and stair-type fishways. Some species of *R. kurodai* migrate between the sea and upstream rivers; therefore, more migrating individuals were observed for *R. kurodai* than for *C. biwae*. This could explain the capture of a larger number of *R. kurodai*.

## Discussion

To restore the continuity of movement for river fish hampered by dams and weirs, adaptive management of fishways is emphasized worldwide. Adaptive management was first described in 1976 under the term "adaptive control" [108]. Originally used in areas related to fisheries resource management, it has also been used in forest and river ecosystem management in Australia and North America [109,110]. The long-term sustainability of natural ecosystems and ecological links is of paramount importance. In scenarios where scientific forecasts are unreliable, adaptive management strategies are employed, aiming to mitigate various natural uncertainties by reducing them over time [111]. To navigate these uncertainties, the integration of adaptive learning and feedback mechanisms is crucial [112]. Adaptive learning consists of active investigation to obtain information efficiently and passive investigation to obtain information from normal investigation [113].

These management techniques, which carefully monitor nature's response to projects and subsequently integrate these observations into project plans, are combined with traditional Japanese knowledge and techniques [114,115]. Management to ensure biodiversity is also underway in Japan, with the enactment of the Nature Restoration Promotion Act (a law enacted by legislators in December 2002). The fishway at the Miyanaka Intake Dam has been appropriately maintained as a civil engineering structure. However, management that focuses on fish in consideration of biodiversity was not implemented. Following the completion of fishway improvements in 2012, a five-year monitoring survey up to 2016 showed that the improvements to the fishway structure were having the desired effect. However, because the habitat and migratory environment of fish may change in the future, monitoring to understand the status of fish migration is ongoing. Continuous monitoring verified that the fishway was functioning as intended and was effective. To adapt without changing the goal [116], it is crucial to develop new hypotheses and implement corresponding measures. The process of repeating this series of processes through trial and error is part of the ongoing adaptive management strategy being implemented at the fishway at the Miyanaka Intake Dam.

A rock-ramp-type fishway was installed at the Miyanaka Intake Dam in 2012, and adaptive management has continued since then, which involved fish catch surveys for monitoring [117]. The burden of this method on the fish and facility managers is heavy; therefore, a survey method utilizing environmental DNA, established in recent years, was adopted. Environmental DNA studies have limitations, one of which is the survival time of the samples. Considering the challenges in sample transportation while ensuring the preservation of samples, we developed an on-site filtration system and a mechanism for freezing and preserving the samples at −20˚C. Environmental DNA surveys (metabarcoding) conducted in 2021 and 2022 identified 30 and 31 species of fish, respectively, of which 24 and 22 species of fish were identified in common from the capture surveys, respectively, based on the sum of those from the three fishways (ice-harbor-type, stair-type, and rock-ramp-type). A comparison of the results of 2021 and 2022 revealed an increasing trend in the environmental DNA results for *P. nudiceps* and *S. biwae*, which increased significantly in the capture survey. Therefore, when the number of individuals that migrated upstream increased, the change was reflected in the environmental DNA survey. Determining the fishway usage status of each fish species in each fishway is

possible by calculating the ratio of the number of environmental DNA reads of the main fish in each fishway and the total fish classified into groups.

A comparison of the results of previous environmental DNA surveys with fish catch surveys showed that the number of species recorded using the environmental DNA surveys tended to be higher than that recorded through fish catches during the same survey period. This discrepancy can arise if only a small number of fish inhabit a specific area, with their detection influenced by factors, such as the flow rate of the river, strength of the current, and behavior of the fish. This pattern is consistent with observations made in numerous studies across various communities and ecosystems [118,119]. The fish recorded at the fishway exit (upstream end of the fishway) through the catch survey were those that had migrated up the fishway. In contrast, the fish confirmed through the environmental DNA survey may have lived upstream of the Miyanaka Intake Dam or potentially migrated upstream along the fishway. Therefore, a comparative study with the results of surveys conducted over the past few years is necessary.

This method used for understanding and monitoring biodiversity presents certain limitations and challenges, such as incomplete recovery of species with very low abundance [120], reliance on methods that need to be further developed in terms of technological advancement and standardization of procedures [121], and issues with abundance quantification, taxonomic assignment, and environmental DNA spatial and temporal dynamics [122]; thus, further optimization is required [123,124]. The results of the catch survey in the fishway included only the types of fish that temporarily reached the trap installed at the upstream end of the fishway. The environmental DNA survey uses the difference between the fishway entrance (downstream) and exit (upstream); therefore, only the types of fish remaining in the fishway were included. However, it is assumed that the samples collected for environmental DNA analysis contain the environmental DNA of fish that lived upstream, in addition to the fish that stayed in the fishway. Therefore, we aimed to confirm whether the frequency of water sampling and the results of the environmental DNA analysis reflected the actual situation.

Environmental DNA is a widespread survey method used to understand the status of fish habitat distribution and reduce the load on fish, and it was used in this study to evaluate the adaptive management of the fishways in the Miyanaka Intake Dam. Environmental DNA analysis was incorporated into the plan as a research method. The surveys in this study are still in the experimental stage of determining the number of fish that have migrated upstream. However, by determining the types of fish using environmental DNA over a long period, it is possible to continuously confirm the presence or absence of changes in fish fauna. The appropriate frequency of water sampling and analysis is 1 week before and after or 15 days before and after. While continuing the survey, which mainly analyzed environmental DNA, the applicability of the analysis was verified by capture surveys conducted every 5 years.

This research presented certain limitations. For example, the water used for analysis in the environmental DNA study was sampled at hourly intervals at the downstream end of the fishway; therefore, the water samples may have contained environmental DNA originating from the following situations: fish living upstream of the Miyanaka Intake Dam, fish staying in the fishway without reaching the net cage at the upstream end of the fishway, and fish caught in net cages. Environmental DNA research methods differ from those used to confirm the biology of fish, and environmental DNA survey results cannot specifically reveal the behavior of the fish, such as whether identified fish were individuals that previously migrated upstream through the fishway, individuals that are currently migrating upstream, or individuals that reside in the fishway.

In this study, the survey compared with the environmental DNA survey targeted fish migrating upstream through the fishway. The hole of the trap for catching fish faced downstream. Considering the viewpoint of the fish, the water sampling point on the downstream side where fish enter the fishway was determined to be the entrance point of the fishway.

However, from a river-based perspective, where environmental DNA flows from upstream, the upstream side of the fishway may be considered as the entry point of the fishway. No survey method observed fish moving down the adjacent fishway when maintenance flow is being released from the spillway gate. Owing to the increased concentration of environmental DNA after spawning [125,126], environmental DNA perturbs the detected fish population. With river flows being continuous and the life history of fish varying, these considerations are crucial for future research on fish descending downstream from dams [127]. Therefore, identifying the target fish and the optimal timing for surveys is essential.

To compare the environmental DNA survey results with the fish capture survey results, the samples in this study were collected at times when the number of captured individuals and species was high based on the results of the capture survey. Environmental DNA analysis was also conducted on samples taken at times when many representative species, such as *P. altivelis*, *T. hakonensis*, *O. platypus*, *C. pollux*, and *R. kurodai*, were captured. Normally, the number of species and populations of fish cannot be determined unless a capture survey is conducted. Therefore, future challenges lie in clarifying the frequency and time of water sampling that will enable both improved accuracy of environmental DNA survey results and improved efficiency of long-term monitoring by analyzing the relationship among various environmental factors, including water temperature and fish migration timing. In addition, given the special situation of monitoring within a fishway rather than within a river, water sampling locations within the fishway should be identified that can more accurately reproduce the local situation.

The impact of fish caught in net cages must also be examined to resolve this issue by collecting water samples, setting times when net cages are not set up, or setting specific days during the survey to capture fish. To compare the results of environmental DNA surveys and fish capture surveys and verify the differences, careful and continuous research must be conducted over a certain number of years.

## Conclusion

We verified whether environmental DNA analysis could be used as a substitute for traditional surveying methods by performing a comparison with the capture survey method using five representative species (*O. platypus*, *T. hakonensis*, *P. altivelis*, *C. biwae*, and *R. kurodai*) around the Miyanaka Intake Dam. The results of environmental DNA analysis for 4 years (2019–2022) and capture surveys for 11 years (2012–2022) were compared. The substitutability of environmental DNA analysis was confirmed based on a comparison period of 1 week to 15 days and according to its tolerance of certain errors, and the proposed method achieved both reduction in the burden on fish and facility managers and a semi-permanent adaptive management strategy for fishways.

Thus, we developed a survey method using a new environmental DNA analysis method for fish passages far from the analysis point. However, environmental DNA research methods are continuously evolving, and trial-and-error methods are applied worldwide. Therefore, we will continue to confirm the applicability of this survey method, which mainly analyzes environmental DNA based on capture surveys and findings obtained every 5 years.

## Acknowledgments

We thank Editage (www.editage.com) for their assistance with English language editing.

## Author Contributions

**Conceptualization:** Masahiko Nakai, Taku Masumoto, Takashi Asaeda.

**Data curation:** Taku Masumoto, Takashi Asaeda, Mizanur Rahman.

**Formal analysis:** Masahiko Nakai, Taku Masumoto, Takashi Asaeda.

**Funding acquisition:** Masahiko Nakai, Taku Masumoto.

**Investigation:** Masahiko Nakai, Taku Masumoto, Takashi Asaeda.

**Methodology:** Masahiko Nakai, Taku Masumoto, Takashi Asaeda, Mizanur Rahman.

**Project administration:** Masahiko Nakai, Taku Masumoto.

**Resources:** Taku Masumoto.

**Software:** Taku Masumoto.

**Supervision:** Masahiko Nakai.

**Validation:** Masahiko Nakai, Taku Masumoto, Takashi Asaeda, Mizanur Rahman.

**Visualization:** Taku Masumoto, Takashi Asaeda, Mizanur Rahman.

**Writing – original draft:** Masahiko Nakai.

**Writing – review & editing:** Masahiko Nakai, Taku Masumoto, Takashi Asaeda, Mizanur Rahman.

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
