## [Decision Letter · Decision Letter 0]

6 Oct 2023

PONE-D-23-20462Improving the efficiency of adaptive management methods in multiple fishways using environmental DNAPLOS ONE

Dear Dr. Masumoto,

Thank you for submitting your manuscript to PLOS ONE. After careful consideration, we feel that it has merit but does not fully meet PLOS ONE’s publication criteria as it currently stands. Therefore, we invite you to submit a revised version of the manuscript that addresses the points raised during the review process.

We look forward to receiving your revised manuscript.

Kind regards,

Petr Heneberg

Academic Editor

PLOS ONE

Journal Requirements:

3. We note that Figures 1, 2 and 3 in your submission contain copyrighted images. All PLOS content is published under the Creative Commons Attribution License (CC BY 4.0), which means that the manuscript, images, and Supporting Information files will be freely available online, and any third party is permitted to access, download, copy, distribute, and use these materials in any way, even commercially, with proper attribution. For more information, see our copyright guidelines: http://journals.plos.org/plosone/s/licenses-and-copyright.

a. You may seek permission from the original copyright holder of Figures 1, 2 and 3 to publish the content specifically under the CC BY 4.0 license. 

Reviewers' comments:

Reviewer's Responses to Questions

**Comments to the Author**

1. Is the manuscript technically sound, and do the data support the conclusions?

Reviewer #1: Partly

Reviewer #2: Yes

2. Has the statistical analysis been performed appropriately and rigorously? 

Reviewer #1: I Don't Know

Reviewer #2: Yes

3. Have the authors made all data underlying the findings in their manuscript fully available?

Reviewer #1: No

Reviewer #2: Yes

4. Is the manuscript presented in an intelligible fashion and written in standard English?

Reviewer #1: No

Reviewer #2: Yes

5. Review Comments to the Author

Reviewer #1: The manuscrispt presents a interesting question and good data. However the lack of objective drive text and clarity made difficult follow and avaliate the manuscript. For instance, the introduction is vey confuse citing several methods and eDNA limitations without a deep discussion on it and just now it cite metabarcoding method, in the previous paragraph the authors are talking about real-time PCR. It is necessary that introduction be more objective. I would suggest around 4 paragraphs is an order like:

1.Dam limit migratory fish movement and isolated populations;

2. Monitoring with traditional methods is expensive and time-consuming;

3. Metabarcoding of eDNA can overcome these limitations but have its own caveats;

4. Goals and hypothesis.

Also, at the introduction the author present a lot about positive controls but not show how it was availiated in the methods, the author even do not cite negative controls. The deposit of the sequences are not mencioned and there are a lack of a lot of information. Was the sequencing paired? Target how many basepairs? How the clean steps (filtering by quality, remotion of primers, treaming, remotion of chimeric sequences, etc) was made? In with similarity OTUs was clustered?

The results are not cleary presented with a many parts apearing more a goal or methods. Because of no clarity in the goal, methods and results I fill not able to availiated the discussion.

I think the manuscript have a great potential if it becames more objective and clear.

Below others minor comments:

Lines 49-50. I did not follow this sentence. Pease, re-write.

Lines 50-51: Not recent years, it is quite old already.

Lines 60-61: Why it is introducing marine environments?

Lines 62-65: This paragraph does not add any useful information.

The introduction could start from line 66.

Lines 85-86: There are studies:

Gelder, J., Benitez, J. P., & Ovidio, M. (2023). Multi-year analysis of the fish colonisation dynamic in three newly installed fishways in medium sized Belgian rivers. Knowledge and Management of Aquatic Ecosystems, 424(12).

Dal Pont, G., Duarte Ritter, C., Agostinis, A. O., Stica, P. V., Horodesky, A., Cozer, N., ... & Pie, M. R. (2021). Monitoring fish communities through environmental DNA metabarcoding in the fish pass system of the second largest hydropower plant in the world. Scientific Reports, 11(1), 23167.

Ritter, C. D., Dal Pont, G., Stica, P. V., Horodesky, A., Cozer, N., Netto, O. S. M., ... & Pie, M. R. (2022). Wanted not, wasted not: Searching for non-target taxa in environmental DNA metabarcoding by-catch. Environmental Advances, 7, 100169.

Line 149: How many litters of water was collected? How? In which deep? How were bags cleaned? Which was the preservation method? How long until the samples to be extracted?

Lines 241-249: No useful information, metabarcoding is already a established technique and much of this information is present in the introduction.

Lines 253-257: Cite Mifish primers developer authors.

Lines 272-273: I did not understand this sentence, clarify.

Table 1: All primers are 5’-3’ oriented? Give the references for each primer used.

Lines 300-306: A lot of information is missing here. Was the sequencing paired? Target how many basepairs? How the clean steps (filtering by quality, remotion of primers, treaming, remotion of chimeric sequences, etc) was made? In with similarity OTUs was clustered?

Lines 345-346: Unnecessary.

Table 2: I did not understand the numbers in parentheses and the column “Fish confirmed at a high frequency”.

Line 378-391: It is goal not a result, remove from here.

Lines 393-412: It is methods.

Lines 414-428: Methods.

Lines 445-45-: I did not follow it.

Reviewer #2: The paper is a pilot application of a coupled eDNA and remote sensing environmental monitoring approach to monitor the impact of a type of sustainable agriculture practice (shaded cocoa plantations), on ecosystem services in Brazilian Amazon. The sustainable agriculture practices of this type, aim at the improvement of the degraded by intensive agriculture ecosystems, while allowing income creation. The work monitors annually the change of land cover, within a five-year time frame. Both methods are highthroughput data producers, making it possible to monitor landscape traits and biodiversity at spatial scales intractable and too costly by traditional ecological methods, and in timeless way. This makes the approach appropriate for environmental management.

The authors give the data in a GitHub link, and detailed presentation of the methods and protocols in the body of the paper, and/or as supplementary material.

The paper has additional value in making steps towards standardisation of the coupled approach for the type of ecosystems it addresses.

ACCURACY ASSESSMENT OF MAPPING PRODUCTS has been an important contribution of this paper having allowed to identify sources of errors in the case of degradation and regeneration and offered valuable lessons for the use of remote sensing. A lesson learnt is that there is need for algorithm parametrization in function of local conditions, so that the method cannot be standardized in a generic manner for all types of ecosystem types and types of change.

The authors deliver new bioindicators more sensitive in detecting change in the short time framework considered. The traditional bioindicators are not sensitive enough to biodiversity change in short time frames. eDNA based new indicators are introduced, with taxon free community structure of arthropods focused indicator and b-diversity. It is rightly pointed out that introducing new indicators in monitoring programs is challenging, before getting broadly accepted and used in standardised ways, since this require thorough reporting and communication.

Some comments/thoughts to the authors:

1. The use of shaded cocoa, is apparently a well-established procedure in Latin America, but still exotic to many people around the world. It may be useful to further explain about it. How long does it take to have both phases established (starting from pasture)? The described procedure resembles an artificially aided ecological succession: (a) annual crops like cassava and maize cultivation, (b) planting bananas and papayas, and (c) parallel native regeneration? (d) cutting bananas and papayas and planting shaded cocoa while native regeneration has progressed (offering shade to shaded cocoa)? It is explained in the text, but it is difficult to conceive the whole succession process. What if the figures of types of vegetation in the eDNA sample collection protocol be included in the paper?

2. To improve knowledge about processes, it would be useful to test the survival of eDNA in different soil types. Some indication do constitute the difference in number of reads between pasture and shaded cocoa, assigned to higher moisture in the latter. Nevertheless experimental evaluation would be very useful.

eDNA under special conditions may survive from days up to years (Tabetlet et al., 2018). eDNA normally has a short lifespan in humid and moist soils, such as the soils in the tropics. On the other hand, eDNA degradation rate is negatively correlated with soil organic carbon and forest soils have the slowest degradation rate, and meadow soils had the greatest stabilization of eDNA (Sirois et al., 2018). In few words, we have no good knowledge about how it goes with insect eDNA lifespan and detectability in soil types sampled in the paper. Also, does the chitin made insect exoskeletons might protect eDNA from degradation?

3. I copy (page 15- On top of Table 3 ):

“The regeneration class presented low user’s and producer’s accuracies and higher levels of uncertainties. The biggest error source was regeneration pixels being committed to the nonregeneration class, i.e., an overestimation of areas where regeneration occurred”.

If my reading is correct (sorry English is not my native language), I understand from the above that regeneration pixels are committed to (=assigned to) the non-regeneration class (erroneously). So that, an underestimation (rather than overestimation) of areas where regeneration occurred should be the case. As written the two expressions are incompatible and we do not know in which of the two the error occurs.

- In any case, it might be written in a more unambiguous way.

4. When is mentioned the bare world "forest", does it always refers to "secondary forest"? "Native forest" is always plainly mentioned? For instance I copy from page 16:

"No Hymenoptera or Lepidoptera were detected in forests. No species were consistently

found to be indicator species"

Here it refers to secondary forest, or to native forest?

6. PLOS authors have the option to publish the peer review history of their article (what does this mean?). If published, this will include your full peer review and any attached files.

Reviewer #1: No

Reviewer #2: No

---

## [Author Response · Author response to Decision Letter 0]

19 Dec 2023

I was satisfied with the constructive discussion with you.

I appreciate your further response.

---

## [Decision Letter · Decision Letter 1]

29 Jan 2024

PONE-D-23-20462R1Improving the efficiency of adaptive management methods in multiple fishways using environmental DNAPLOS ONE

Dear Dr. Masumoto,

Thank you for submitting your manuscript to PLOS ONE. After careful consideration, we feel that it has merit but does not fully meet PLOS ONE’s publication criteria as it currently stands. Therefore, we invite you to submit a revised version of the manuscript that addresses the points raised during the review process.

We look forward to receiving your revised manuscript.

Kind regards,

Petr Heneberg

Academic Editor

PLOS ONE

Reviewers' comments:

Reviewer's Responses to Questions

**Comments to the Author**

1. If the authors have adequately addressed your comments raised in a previous round of review and you feel that this manuscript is now acceptable for publication, you may indicate that here to bypass the “Comments to the Author” section, enter your conflict of interest statement in the “Confidential to Editor” section, and submit your "Accept" recommendation.

Reviewer #2: All comments have been addressed

2. Is the manuscript technically sound, and do the data support the conclusions?

Reviewer #2: Yes

3. Has the statistical analysis been performed appropriately and rigorously? 

Reviewer #2: Yes

4. Have the authors made all data underlying the findings in their manuscript fully available?

Reviewer #2: Yes

5. Is the manuscript presented in an intelligible fashion and written in standard English?

Reviewer #2: Yes

6. Review Comments to the Author

Reviewer #2: General comments:

Towards standardization/benchmarking of eDNA approach monitoring river fish populations, against capture survey data:

The paper addresses in a clear way an interesting question and a process of developing of a method by means of eDNA analysis. The results are also interesting. In our opinion it shows that eDNA methodology for monitoring populations and communities, despite some principles of general application, should be custom designed for specific conditions and the questions. On the other hand, there are some points in the paper, that are not so easy to follow.

Some key findings of the paper

(that are not easy to find somewhere in a very concise way):

There are two parameters most essential to describe the structure of a biological community in ecology: (a) the identification of species composition in terms of presence absence, (b) the abundance of each one. The paper addresses both parts, since they are necessary for effective monitoring:

Presence-absence: The identification of species by eDNA performs very well, and follows well, the effect of the type of fishway on the fish species crossing each one, in good agreement with the capture survey.

Quantification (relative abundance): The method, also offers some quantitative capacity shown in two ways (i) in the relative abundance of each fish species passing each fishway vs all others, which corresponds well with capture survey, (ii) in the comparison within species, between years of the eDNA reads with a significant increase in numbers for some species for both capture survey and eDNA reads from 2021 to 2022.

On the other hand, I have had some difficulties to follow the reasoning about the effect of time distance from sampling time point on eDNA based estimations. I have no suggestions for improving it and at the end, I got the picture.

Introduction

Some fish biology would be helpful. Are the fish species identified anadromous, or freshwater only? The word anadromous is not used in any place in the text (it can found in reference 36 only). Instead there is the word “migratory”. I do not know if these two are considered synonymous here). Are there any fish species that are exclusively freshwater that are migratory? On the other hand, the authors considered the biology and behavior of different fish species later in the discussion, to interpret some results on eDNA. This sheds convincing light on the results and shows a careful work. These two, could be better linked, with a small paragraph in the introduction.

The word “adaptive management” is in the title, in the short title, in the key words, twice in the abstract and four times in the rest of the paper. It is not a trivial thing, so that most people do not know what it is. There should be few lines in this in the introduction on its general connotation and on its specific to the damns and fishways. This could be in 45, where it is first introduced in the paper. There is a very good present of the term in Wikipedia, and certainly there are good references in scientific papers.

The word “conservation” is comprised in the key words, but it is hardly mentioned (just once in line 82) or the value of the present work for species conservation and management of threatened or endangered species is developed (vs management for fisheries purposes for instance), although it does appear in the title of 8 papers in the references. I copy any expression from the abstract of the paper DOI: 10.7541/2018.152, (1st that I found by googling): “The functions of fishways gradually transferred from conservation of economic stocks to overall biodiversity conservation…”

Methods

The three types of fishway that are designed to facilitate the movements of fish having different biology and behaviors, offer a very good setting to test the effectiveness of eDNA in terms of specificity and detection power, under different conditions of water flow. It is reported in line 135-136 (“the most distinctive feature of the Miyanaka Intake Dam is that

the fish fauna is different in each of the three fishways”), and in my view, this should be stated in a stronger way even earlier in the paper.

Discussion

The discussion is very long, and it comprises results, that should be placed in the results section. This is also the case for some figures which are places in the Discussion section. Some methodological points become clear, only in the discussion and not in the Materials and Methods section. So, I would suggest some reshaping of the paper, not on the science grounds, but on clarity of presentation.

Some more specific comments:

1. Lines 136 – 137: “This comparison had to be characterized”.

It might be interesting something like: «This offers a fine-grained system to benchmark eDNA against capture surveys».

Despite the material procured in the paper, like plans and photos, it is not very easy to figure out the setting and how the stations are positioned, although the text helps quite a lot. One difficulty for instance comes from naming “Exit of fish ways” the upstream part of the installation, where the flow of the river comes in, so by “river thinking”, not “fish thinking” I would expect it to be named “entrance of fish ways”. The authors are experts on the matter and may make some effort on this. It is clear at the end of the paper, but it requires some extra effort. Here also comes the need to have previously (introduction) write something in the biology of the species: which of the two is the impact of a damn of riverine fish populations: (a) incurring population fragmentation on exclusively freshwater species using the whole length of the river the whole year, and therefore decreasing connectivity, gene flow, and increasing genetic drift, or (b) preventing anadromous fish species to reach the upstream reproduction grounds? The paper sems to address only the later (migratory fish) and considers only the upstream migration of mature fish, not the downstream of juveniles. Depending on what part of migratory fish cycle we address, the entrance and exit of fishway is different.

2. Lines 183-184: “Environmental 184 DNA Sampling and Experiment Manual Version 2.1 (published on April 25, 2019).

I consider the care of authors for standardisation of methods, which makes studies comparable, a very important contribution. The authors indirectly state it the paper (Lines 249-252) and it is great to know that there is in Japan an “environmental DNA Methods Standardization Committee”. Still they should not shy away from stressing their huge effort to standardise eDNA in such a complex system, as a major contribution to a global science against global challenges.

3. Lines 243-249 on explaining the “DNA metabarcoding method”:

I think this can be written in a simpler way, or not explained at all and simply give two good references, including the reference that you are giving already below [94].

Here I give a try to an expression (and you can find much better in the literature):

e.g. DNA metabarcoding is a Next Generation Sequencing based method applied on environmental samples, allowing to characterize the species composition of whole communities by using marker genes [references].

4. Line 251: “Environmental DNA Methods Standardization Committee”.

It is very positive that there is such a committee; it is expected that their standardised protocols be readily available and easily findable and accessible).

5. Lines 273 : “repeating the 1st PCR eight times”.

May it is good to explain a reason for so many repeated amplifications...or give a reference if this has been addressed and explained in other work.

6. Lines 280-281: “However, if the fish are less active, it should be increased to a maximum of 40 281 cycles to recover a sufficient amount of DNA”.

This is a very interesting observation. Although this sounds common sense, it is worth to be discussed as function of own data, or with a pertinent reference, if this has been shown other works. I have already commented positively elsewhere on the paper’s use of biology and behaviour to interpret data.

7. Lines 289-290: “U is a primer suitable for fish in general, U2 is specific to sea bream, and E-v2 is specific 290 to cartilaginous fish”.

Which of the above species is the one named "sea bream" here? If you google sea bream, you will find specifies the Mediterranean fish species "Spatus aurata". Also, which of the studied species is cartilaginous? I personally ignored that there are cartilaginous freshwater fish; it is maybe interesting to know for all the readers.

8. Lines 294-297: “A (adenine), T (thymine), G (guanine), and C (cytosine) in the table are the four types of bases that constitute DNA. In double-standard DNA, A and T and G and C have complementarity pairing. “NNNNNN” means one of A, T, G, or C”.

I think that all this should be deleted as unnecessary information.

9. Lines 355-356 – Table 2: “DNA of detected species”.

I think it is more precise, or clearer the expression: “number of species detected by eDNA”.

10. Lines 487-490: “Environmental DNA surveys (metabarcoding) conducted in 2021 and 2022 identified 30 and 31 species of fish, respectively, of which 24 and 22 species of fish, respectively, were identified in common from the capture surveys as the sum of those from the three fishways (ice-harbor-type, stair-type, and rock ramp-type)”.

We learn later in the text, that the total number of species captured over 2021&2022 is 37 species and that this is also the number of species identified by eDNA, and that they are not the same, but that there is a very good overlap. I think that it is good to bring at this point the information, so that the text that follows is an easier read: "37 species were identified in the fish catch surveys from 2021 to 2022".

11. Lines 497-499: “Comparing the results of previous environmental DNA surveys with fish catch surveys, the number of species recorded using the environmental DNA surveys tended to be higher than those recorded through fish catches during the same survey period”.

Could be added:

…which is a more general trend as shown by most studies across communities and ecosystems [giving also some REFs]

7. PLOS authors have the option to publish the peer review history of their article (what does this mean?). If published, this will include your full peer review and any attached files.

Reviewer #2: No

---

## [Author Response · Author response to Decision Letter 1]

7 Mar 2024

I greatly appreciate your constructive and thoughtful comments.

---

## [Editor Report · Decision Letter 2]

13 Mar 2024

Improving the efficiency of adaptive management methods in multiple fishways using environmental DNA

PONE-D-23-20462R2

Dear Dr. Masumoto,

We’re pleased to inform you that your manuscript has been judged scientifically suitable for publication and will be formally accepted for publication once it meets all outstanding technical requirements.

Kind regards,

Petr Heneberg

Academic Editor

PLOS ONE

---

## [Editor Report · Acceptance letter]

20 Mar 2024

PONE-D-23-20462R2 

PLOS ONE

Dear Dr. Masumoto, 

I'm pleased to inform you that your manuscript has been deemed suitable for publication in PLOS ONE. Congratulations! Your manuscript is now being handed over to our production team.

Kind regards, 

on behalf of

Dr. Petr Heneberg 

Academic Editor

PLOS ONE